# Bridging of nucleosome-proximal DNA double-strand breaks by PARP2 enhances its interaction with HPF1

Guillaume Gaullier[1,2¤], Genevieve Roberts[1], Uma M. Muthurajan[1,2], Samuel Bowerman[1,2], Johannes Rudolph[1,2], Jyothi Mahadevan[1], Asmita Jha[1], Purushka S. Rae[1], Karolin Luger[1,2]*

1 Department of Biochemistry, University of Colorado Boulder, Boulder, CO, United States of America,
2 Howard Hughes Medical Institute, University of Colorado Boulder, Boulder, CO, United States of America

¤ Current address: Department of Cell and Molecular Biology, Science for Life Laboratory, Uppsala University, Uppsala, Sweden
* karolin.luger@colorado.edu

**Data Availability Statement:** The datasets produced in this study are available in the following databases: • Cryo-EM map: EMDB EMD-20864

## Abstract

Poly(ADP-ribose) Polymerase 2 (PARP2) is one of three DNA-dependent PARPs involved in the detection of DNA damage. Upon binding to DNA double-strand breaks, PARP2 uses nicotinamide adenine dinucleotide to synthesize poly(ADP-ribose) (PAR) onto itself and other proteins, including histones. PAR chains in turn promote the DNA damage response by recruiting downstream repair factors. These early steps of DNA damage signaling are relevant for understanding how genome integrity is maintained and how their failure leads to genome instability or cancer. There is no structural information on DNA double-strand break detection in the context of chromatin. Here we present a cryo-EM structure of two nucleosomes bridged by human PARP2 and confirm that PARP2 bridges DNA ends in the context of nucleosomes bearing short linker DNA. We demonstrate that the conformation of PARP2 bound to damaged chromatin provides a binding platform for the regulatory protein Histone PARylation Factor 1 (HPF1), and that the resulting HPF1•PARP2•nucleosome complex is enzymatically active. Our results contribute to a structural view of the early steps of the DNA damage response in chromatin.

## Introduction

DNA double-strand breaks (DSBs) are caused by ionizing radiation, chemicals or replication stress and are the most cytotoxic of all DNA lesions. Their occurrence elicits a very rapid cellular response called the DNA damage response (DDR), leading to cell-cycle arrest until the break is repaired or until the cell is directed to apoptosis. Understanding the DDR is critical to optimize cancer therapies that inflict an overwhelming amount of DNA damage to tumor cells. Inhibitors of the DDR are indeed used in combination with radiotherapy to lower the radiation dose required to efficiently kill cancer cells [1–3].

Upon DNA damage, many nuclear proteins around the damage site rapidly accumulate a post-translational modification called poly(ADP-ribose) (PAR), which constitutes the earliest

(https://www.ebi.ac.uk/pdbe/entry/emdb/EMD-20864) • Cryo-EM raw movies and particle coordinates: EMPIAR EMPIAR-10336 (https://www.ebi.ac.uk/pdbe/emdb/empiar/entry/10336) • Atomic model: PDB 6USJ (https://www.rcsb.org/structure/6USJ) • Gel images, fluorescence polarization and FRET binding data, analytical size exclusion chromatograms, SEC-MALS data, differential scanning fluorimetry: Zenodo (https://doi.org/10.5281/zenodo.3519436)

**Funding:** R01 CA218255-01 Howard Hughes Medical Institute T32 GM008759 20POST35211059.

**Competing interests:** The authors have declared no competing interest.

signaling event in the DDR and functions as an anchor to recruit a variety of downstream signaling and repair factors to the sites of DNA breaks [4]. The vast majority of PAR is synthesized from nicotinamide-adenine dinucleotide ($NAD^+$) by two DNA-dependent poly(ADP-ribose) polymerases (PARPs), PARP1 and PARP2, which belong to a family of 18 proteins containing a conserved ADP-ribosyl transferase (ART) domain [5]. PARP2 was discovered as the enzyme responsible for the residual amount of PAR detected in *Parp1*$^{-/-}$ cells [6]. PARP2 also accumulates at sites of DNA damage, although slower than PARP1 [7, 8]. PARP2, in addition to polymerizing PAR, establishes more branching in the PAR chains already synthesized by PARP1 [9].

*In vitro* studies showed that PARP1 and PARP2 carry out automodification [6, 10], i.e. they attach PAR chains to themselves. It was demonstrated more recently that PARylated PARP1 and PARP2 dissociate from DNA as the PAR chains grow longer, likely due to electrostatic repulsion between the DNA backbone and the PAR phosphate groups [11]. This raises the question of how PAR chains can efficiently signal the location of a double-strand break if the protein they are attached to eventually dissociates from DNA. Proteomics studies have established that many proteins become PARylated when treating cells with DNA damaging agents [12, 13], including PARP1 and PARP2 themselves (auto-modification) [14], and histones [15]. PARylation of histones ensures that PAR chains remain close to the damage site even after auto-modified PARP1 and PARP2 dissociate.

PARylation reactions reconstituted *in vitro* from purified recombinant components yielded auto-modified PARP1 and PARP2, and also PARylation in *trans* of other substrates to varying degrees, including core and linker histones. Reactions using nucleosomes prepared from native chromatin could achieve a higher degree of histone PARylation [16, 17], suggesting that reactions reconstituted with recombinant components might have been depleted of a key factor. The recently discovered Histone PARylation Factor 1 (HPF1) modulates the enzymatic activity of PARP1 and PARP2 by redirecting their specificity from Asp and Glu to Ser amino acid side chains, and by redirecting both enzymes to more efficiently PARylate histones in *trans* (in addition to their auto-modification in *cis*) [18]. This has been demonstrated *in vivo* with HPF1 knockouts, and *in vitro* with purified recombinant components where serine PARylation depends on HPF1 [19].

Domains of PARP1 and PARP2 bound to DNA breaks have been structurally characterized [20–24], but no structural information is available in the context of chromatin. GST pull-down experiments demonstrated that HPF1 interacts with the catalytic domain of PARP1 and PARP2 [18], but the binding affinity of HPF1 for full-length PARP1 and PARP2 in their conformation bound to DNA damage has not been quantified. To better understand the molecular mechanism of chromatin ADP-ribosylation, we determined a cryo-EM structure of PARP2 bound to a nucleosome mimicking a double-strand break in chromatin and biochemically characterized the interactions within the enzymatically active HPF1 –PARP2 –nucleosome ternary complex.

## Results

### A Q112R/F113D mutant of human PARP2 yields homogeneous single particles suitable for cryo-EM

In order to prepare a homogeneous complex for structure determination, we performed electrophoretic mobility shift assays (EMSA) of a 165 bp nucleosome (Nuc165) with increasing molar equivalents of wild-type human PARP2. All molar ratios tested resulted in a mixture of species, apparent as a ladder in EMSA (Fig 1A, left lanes), and only yielded aggregates not amenable to analysis by cryo-EM (S1A Fig). We serendipitously discovered a mutant of PARP2,

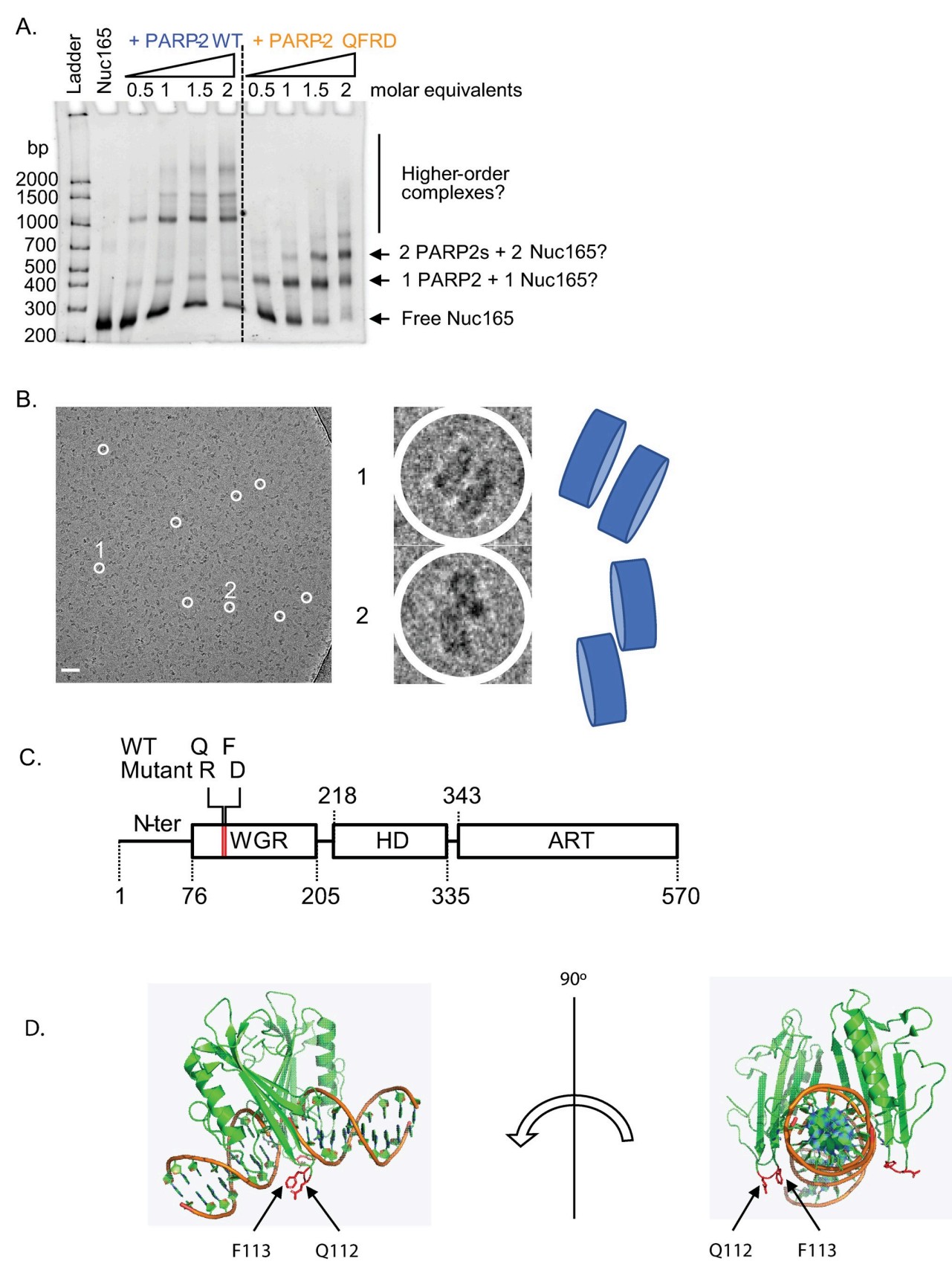

A.

Ladder Nuc165 + PARP2 WT + PARP2 QFRD

0.5 1 1.5 2 | 0.5 1 1.5 2  molar equivalents

bp

2000
1500
1000
700
500
400
300
200

Higher-order complexes?

← 2 PARP2s + 2 Nuc165?
← 1 PARP2 + 1 Nuc165?

← Free Nuc165

B.

C.

WT    Q  F
Mutant R  D

218        343

N-ter        WGR        HD        ART

1    76        205        335        570

D.

90°

F113  Q112        Q112  F113

**Fig 1. A double point mutant of PARP2 forms a homogeneous complex with Nuc165, suitable for cryo-EM. A:** Electrophoretic mobility shift assay of Nuc165 (1.4 μM) with increasing molar ratios of PARP2-WT or PARP2-QFRD (ethidium bromide staining). **B:** Cryo-electron micrograph of the PARP2-QFRD–Nuc165 complex. Enlarged views of two particles are shown (1 and 2). Scale bar: 50 nm. Cartoon representations of the two particles is presented alongside the enlarged views. **C:** Domain structure of human PARP2 and location of the Q112R and F113D double point mutation in the WGR domain. **D:** Location of the Q112 and F113 residues (drawn as red sticks) in the context of the PARP2-WGR–DNA complex crystal structure (PDB entry 6F5B).

Q112R, F113D (referred to as PARP2-QFRD), originally designed based on HDX-MS data showing protection of a peptide encompassing residues Q112 and F113 of PARP2 when mixed with PARP1. The mutations were designed to be solvent-exposed and to change drastically the properties of amino acids at these positions without changing the protein's isoelectric point). We tested the binding of PARP2-QFRD to the same nucleosomes. In contrast to the wild type protein, the double point mutant PARP2-QFRD does not form these slow-migrating PARP2 – Nuc165 complexes and forms better-defined shifted complexes (Fig 1A, right lanes). Most importantly, the complex formed with PARP2-QFRD and Nuc165 at the same molar ratio and under the same conditions tested for PARP2-WT yielded homogeneous single particles suitable for cryo-EM analysis (Fig 1B). Close inspection of these particles suggested significant nucleosome dimerization induced by PARP2 (Fig 1B, enlarged views).

Besides its apparently better-defined nucleosome-binding properties (Fig 1A), PARP2-QFRD is otherwise indistinguishable from PARP2-WT *in vitro*: it is properly folded, with a $T_m$ value within 1˚C of that of PARP2-WT (S1B Fig), it binds a 5'-phosphorylated 18 bp DNA (p18mer) with the same affinity as PARP2-WT (S1C Fig and Table 1), it binds Nuc165 with the same affinity as PARP2-WT (S1D Fig and Table 1) and it is enzymatically fully active (S1E Fig) with a $K_{act}$ for DNA close to that of PARP2-WT (S1F Fig). Moreover, its $K_D$ for p18mer has a similar pronounced ionic strength dependence as that of PARP2-WT (S1G and S1H Fig and S1 Table), indicating the same predominant electrostatic contribution to DNA binding. We could also replicate the $K_D$ of ~240 nM at higher salt concentrations reported previously [11] for both PARP2-WT (S1I Fig) and PARP2-QFRD (S1J Fig).

The crystal structure of the WGR domain of PARP2 bound to 5'-phosphorylated DNA (PDB entry 6F5B) [24] shows that the two mutated residues (Q112 and F113) are located in a loop that is close to the DNA, but not directly involved in DNA binding (Fig 1C and 1D). A conservation analysis indicates that these two residues are not conserved among PARP2 orthologues (S1K Fig), in contrast with residues W138, G139 and R140 that form a conserved WGR

**Table 1. $K_D$ values determined in this study.**

| Interaction (titrant / probe) | $K_D$ (nM) | Hill coefficient | Number of replicates | Method | Figure |
|---|---|---|---|---|---|
| PARP2-WT / p18mer_fluorescein | 11.1 ± 0.4 | 1.2 ± 0.0 | 3 | FP | S1C |
| PARP2-QFRD / p18mer_fluorescein | 17.1 ± 1.8 | 0.9 ± 0.1 | 3 | FP | S1C |
| PARP2-WT / Nuc165_Alexa488 | 75.9 ± 2.8 | 1.9 ± 0.1 | 3 | FP | S1D |
| PARP2-QFRD / Nuc165_Alexa488 | 65.6 ± 2.8 | 1.9 ± 0.1 | 3 | FP | S1D |
| HPF1 / Nuc165_Alexa488 | 1 451.8 ± 148.2 | 1.3 ± 0.1 | 3 | FP | 5A |
| HPF1_Alexa647 / PARP2-WT_Alexa488 | > 3 000 | 1.1 ± 0.0 | 3 | FRET | S5A |
| HPF1_Alexa647 / preformed complex PARP2-WT_Alexa488 + Nuc165 | 278.9 ± 16.6 | 1.7 ± 0.2 | 3 | FRET | S5A |
| HPF1_Alexa647 / PARP2-QFRD_Alexa488 | > 10 000 | 1.0 ± 0.1 | 3 | FRET | 5B |
| HPF1_Alexa647 / preformed complex PARP2-QFRD_Alexa488 + Nuc165 | 701.8 ± 38.5 | 1.3 ± 0.1 | 3 | FRET | 5B |

Reported values of $K_D$ and Hill coefficient are the mean and standard error of the mean from three independent measurements.

motif after which the WGR domain was named, and residue Y188 involved in specific recognition of the terminal 5'-phosphate at a DNA break (PDB entry 6F5B) [24]. However, the Q112 and F113 residues are involved in packing contacts in the crystal lattice (S1L Fig), suggesting that they may contribute to the self-aggregation propensity of PARP2-WT bound to Nuc165 that we observed in EMSA (Fig 1C) and cryo-EM (S1A Fig).

For cryo-EM studies, based on this extensive characterization, we decided to use low salt concentrations (no higher than 50 mM NaCl) and PARP2-QFRD to ensure high-affinity binding of PARP2 to Nuc165 and formation of homogeneous particles.

## PARP2-QFRD bridges nucleosomes by binding to linker DNA ends

Initial cryo-EM images of the PARP2-QFRD•Nuc165 complex indicated nucleosome dimers (Fig 1B, enlarged views). Molecular weights measured by SEC-MALS of complexes formed in solution at different molar ratios of PARP2-QFRD to Nuc165 are consistent with a stoichiometry of 2 PARP2-QFRD and 2 Nuc165 (Fig 2, S2 Fig and Table 2), in agreement with the previously published stoichiometry of PARP2 bound to DNA ends [24]. Neither Nuc165 (at 4.61 μM) nor PARP2-QFRD (at 30 μM) dimerize by themselves (Fig 2A and S2A Fig, respectively). The molar ratio of PARP2-QFRD to Nuc165 determines the relative amounts of 1:1 and 2:2 PARP2-QFRD•Nuc165 complexes. With sub-stoichiometric amounts of PARP2-QFRD, a 1:1 complex is seen along with free nucleosomes (S2B Fig). When mixing the two components at a 1:1 molar ratio, the resulting complex is distributed into roughly equal populations of the 1:1 and 2:2 stoichiometries (Fig 2B). The molecular weight of the peak at 11.8 mL is closer to a 1:1 complex even though the elution volume itself could also reflect unbound nucleosomes (Fig 2B). The 2 Nuc165: 2 PARP2-QFRD complex displays a molecular weight and elution volume close to that of a free dinucleosome indicating the bridging of two mononucleosomes by PARP2 (Fig 2C and 2D). Only at 2 molar equivalents of PARP2-QFRD versus Nuc165 did we observe a majority of bridged nucleosome dimers (S2C Fig), consistent with the $K_D$ of 65.6 ± 2.8 nM (S1D Fig and Table 1) predicting that full saturation of 4.5 μM Nuc165 requires 2 molar equivalents of PARP2 (Eq 1). We therefore prepared all samples for cryo-EM by mixing Nuc165 at low micromolar concentrations with 2 molar equivalents of PARP2-QFRD.

We determined a cryo-EM structure of the PARP2-QFRD•Nuc165 complex at 10.5 Å overall resolution (Fig 3, S3 Fig, Table 3), with local resolution ranging from 8.4 to 20.4 Å (S3C Fig). In an attempt to improve resolution, we collected a second, larger dataset on a different microscope (from a different grid prepared with a different batch of complex). Resolution did not improve, but this independent 3D reconstruction yielded the same final map (S3I–S3N Fig), which superimposes well onto the first map (Fig 3A and S3H Fig) with a real-space correlation coefficient of 0.97 (see Methods). We only discuss the map obtained from the first dataset (Fig 3A and S3H Fig) because it originates from particles that sample more orientations than the particles of the second dataset (compare S3E and S3M Fig). Our cryo-EM map reveals a nucleosome dimer, with additional density in the region of linker DNA by which the two nucleosomes are bridged (Fig 3A). This additional density is well explained by a dimer of the WGR domain of PARP2 (Fig 3B) in the conformation present in the crystal structure of two DNA fragments bridged by the WGR domain of PARP2 (PDB entry 6F5B) [24]. Superimposing a map calculated from PDB 6F5B at 10.5 Å resolution (at a contour level of 0.1) to our experimental map (at a contour level of 0.003) yields a real-space correlation coefficient of 0.92, indicating excellent agreement between our map and PDB entry 6F5B. As shown previously, DNA bridging is stabilized by protein-DNA interactions between the two copies of the WGR domain and the four DNA strands (including interaction of residue Y188 with the

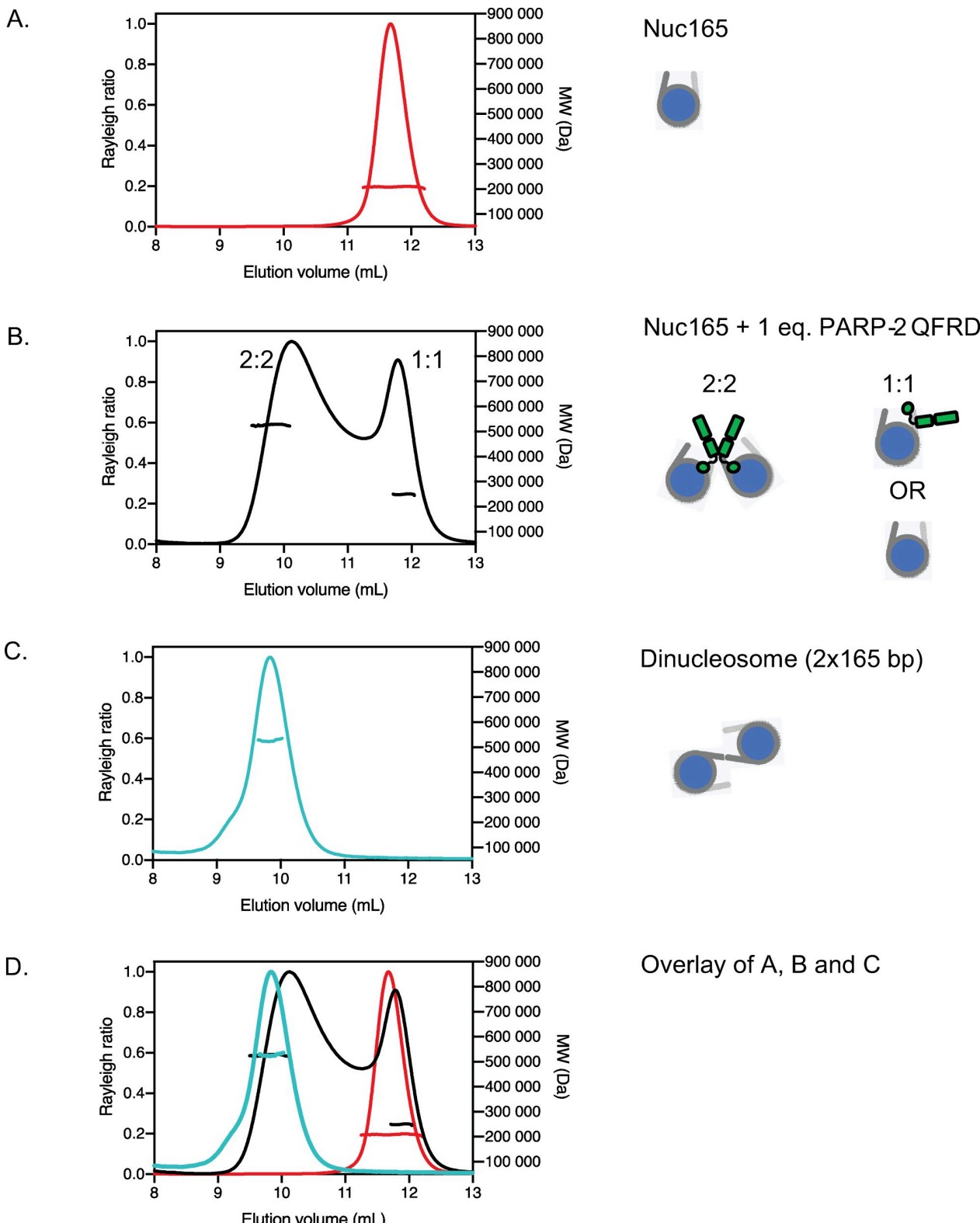

**Fig 2. Stoichiometry of the PARP2-QFRD–Nuc165 complex.** Size exclusion chromatograms and experimental molecular weights determined by SEC-MALS. All molecular weights are listed in Table 2. Stoichiometries consistent with the experimental molecular weights are depicted as cartoons. **A:** Nuc165. **B:** Nuc165 + 1 molar equivalent of PARP2-QFRD. **C:** Dinucleosome 2x165 bp. **D:** Overlay of chromatograms in panels A, B and C.

**Table 2. Molecular weights determined by SEC-MALS.**

| Sample | Figure | Peak (mL) | Stoichiometry Nucleosome | Stoichiometry PARP2 | MW from sequence (Da) | MW from SEC-MALS (Da) | Discrepancy (%) |
|---|---|---|---|---|---|---|---|
| PARP2-QFRD | S2A | 14.5 | 0 | 1 | 66 964.43 | 80 970 | 20.91 |
| Nuc165 | 2A | 11.7 | 1 | 0 | 210 515.06 | 208 600 | -0.91 |
| dinucleosome 2x165 | 2C | 9.9 | 1 | 0 | 456 076.12 | 526 800 | 15.51 |
| Nuc165 + 0.5 eq. PARP2-QFRD | S2B | 11.8 | 1 | 0 | 210 515.06 | 242 200 | 15.05 |
| | | 10.9 | 1 | 2 | 344 443.92 | 346 200 | 0.51 |
| Nuc165 + 1 eq. PARP2-QFRD | 2B | 11.8 | 1 | 1 | 277 479.49 | 249 400 | -10.12 |
| | | 10.2 | 2 | 2 | 554 958.98 | 526 500 | -5.13 |
| Nuc165+ 2 eq. PARP2-QFRD | S2C | 11.0 | 1 | 2 | 344 443.92 | 308 200 | -10.52 |
| | | 9.5 | 2 | 3 | 621 923.41 | 634 600 | 2.04 |

terminal 5'-phosphate at the break site), but the two copies of the WGR domain of PARP2 do not establish protein-protein interactions [24].

Our map does not show density for either the flexible N-terminal region or the catalytic domain of PARP2-QFRD, even though we prepared the complex with full-length PARP2-QFRD. The flexible N-terminal region of PARP2-QFRD (residues 1 to 76, Fig 1C) likely adopts a variety of conformations even in its DNA-bound state, in a manner similar to histone tails [25, 26]. Because image alignment is driven by the two nucleosomes, this intrinsic conformational heterogeneity of the N-terminus of PARP2-QFRD is impossible to classify and resolve in our resolution range. Although larger than the WGR domain, the catalytic domain of PARP2-QFRD was not visible in any of the 2D class averages and 3D reconstructions (S3F, S3G and S3H Fig). SDS-PAGE analysis of samples after grid preparation did not show any sign of PARP2 degradation (S6A Fig, compare to S7A Fig showing PARP2 at its final step of purification). The 12-residue linker between the WGR and catalytic domains may be flexible enough to allow movement of the catalytic domain relative to the more rigid nucleosome dimer, thereby blurring it in 2D class averages because it is too small (relative to the two nucleosomes) to drive image alignment and classification. Missing protein domains in a cryo-EM maps with no evidence of protein degradation is not uncommon (e.g. [27]), and it is now well appreciated that exposure to the air-water interface can lead to denaturation of protein domains [28–30].

Of note, the relative orientation of the two nucleosomes in our set of particles is remarkably uniform (S3H Fig), even though the two WGR domains do not contact each other directly [24]. This is in contrast with previously reported 2D class averages and 3D reconstructions of nucleosome dimers in absence of a binding factor, in which the nucleosomes weakly associate through interactions in *trans* between histone tails and DNA and adopt a variety of relative orientations at 150 mM NaCl [31]. The lower salt concentration in our experimental conditions (5 mM NaCl in the cryo-EM samples that yielded the reconstruction shown in Fig 3) should be permissive to these weak inter-nucleosome interactions, yet we only observe a single relative orientation between the two nucleosomes promoted by the specific and high-affinity interaction between PARP2-QFRD and DNA ends. The proximity of the two bridged nucleosomes may also facilitate inter-nucleosome interactions between the tail of H4 of one nucleosome and the DNA of the other nucleosome, potentially also stabilizing their relative orientation. The relative orientation of the two nucleosomes in our map is also different compared to what is observed in another recent cryo-EM structure of the PARP2•nucleosome

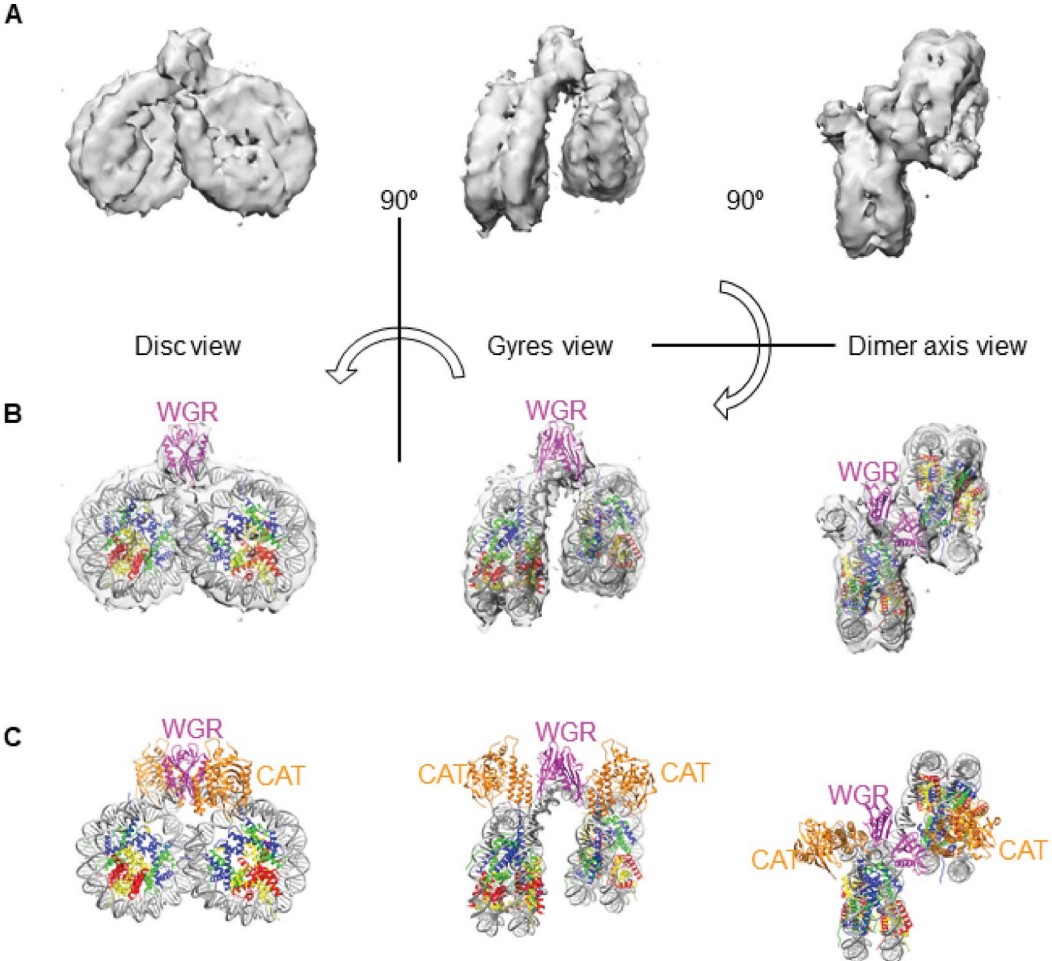

**Fig 3. Cryo-EM structure of the PARP2-QFRD–Nuc165 complex. A:** Single-particle cryo-EM 3D reconstruction of the PARP2-QFRD–Nuc165 complex (contour level of 0.005). **B:** Cryo-EM map with fitted model of two human 601 nucleosomes with 8 bp linker DNA bridged by the WGR domain of PARP2-QFRD (see Methods for details of model building and refinement). Histone H3 is shown in blue, H4 in green, H2A in yellow, H2B in red, DNA in grey and PARP2 WGR domain in magenta. **C:** Model of the PARP2-QFRD–Nuc165 complex, with same color code as in B. A possible location of the catalytic domain (shown in orange) was inferred from structural superimpositions (see Methods for details). The disordered N-terminus region of PARP2 was not modeled.

complex that was released while this manuscript was in revision [32], possibly due to the differences in linker DNA length or to steric constraints imposed by the presence of HPF1.

To provide insights into the structure of full-length PARP2 bound to nucleosomes, we inferred the location and orientation of its catalytic domain. We first superimposed the WGR domain of PARP1 from the crystal structure of PARP1 bound to DNA (PDB entry 4DQY) [21] onto the WGR domain of PARP2-QFRD in our model (RMSD = 0.92 Å between 78 C$\alpha$ pairs). Then we superimposed the catalytic domain of PARP2 (PDB entry 4ZZX) [33] onto the catalytic domain of PARP1 in 4DQY (RMSD = 0.82 Å between 263 C$\alpha$ pairs), and finally removed PARP1 from the model. These two superpositions provide a plausible location and orientation of the catalytic domain of PARP2 in our complex (Fig 3C). The resulting model of two copies of full-length PARP2 bridging two copies of Nuc165 does not generate any steric clashes and represents a physically realistic conformation of the complex. In this model, the longest distance between the N-terminus of the globular domain of H3 and the catalytic

**Table 3. Cryo-EM data collection, processing and refinement statistics.**

| | Dataset 1 | Dataset 2 |
|---|---|---|
| | **PARP2-QFRD/Nuc165** | **PARP2-QFRD/Nuc165** |
| **Data collection 3D reconstruction** | | |
| Facility | Boulder EM Services | Janelia |
| Microscope / detector | Tecnai F30 / K2 | Titan Krios / K2 |
| Acceleration voltage (kV) | 300 | 300 |
| Magnification | 31 000x | 22 500x |
| Super-resolution mode | no | yes |
| Physical pixel size (Å/px) | 1.271 | 1.31 |
| Nominal defocus range (μm) | -1 to -2.5 | -1 to -2.5 |
| Electron dose rate ($e^-$/px/s) | 17.83 | 10 |
| Exposure time (s) | 4 | 8 |
| Total electron dose ($e^-$/Å$^2$) | 56.10 | 58 |
| Number of movie frames | 40 | 50 |
| Dose per frame ($e^-$/Å$^2$) | 1.4025 | 1.16 |
| Number of movies collected | 632 | 2 355 |
| Initial number of particles | 226 250 | 984 171 |
| Number of particles for 3D classification | 168 765 | 983 946 |
| Symmetry | C1 | C1 |
| Number of particles for 3D refinement | 27 889 | 16 304 |
| Resolution at FSC = 0.143 (masked / unmasked) (Å) | 10.5 / 11.6 | 10.5 / 12.3 |
| B factor for map sharpening (Å$^2$) | -1 127.02 | -833.594 |
| **Model refinement** | | |
| Chains | 22 | |
| Atoms | 26 958 | |
| Amino acid residues | 1735 | |
| Nucleotides | 638 | |
| Water molecules | 0 | |
| Bond length RMSD (Å) (number of outliers $> 4\sigma$) | 0.015 (1) | |
| Bond angle RMSD (˚) (number of outliers $> 4\sigma$) | 1.938 (187) | |
| MolProbity score | 0.56 | |
| Clash score | 0.14 | |
| Ramachandran outliers (%) | 0.00 | |
| Ramachandran allowed (%) | 1.59 | |
| Ramachandran favored (%) | 98.41 | |
| Rotamer outliers (%) | 0.21 | |
| Cβ outliers (%) | 0.00 | |
| Peptide plane outliers (%) Cis proline / general | 0.00 / 0.00 | |
| Peptide plane outliers (%) Twisted proline / general | 0.00 / 0.00 | |
| CaBLAM outliers (%) | 0.78 | |
| Protein B factors (Å$^2$) min/max /mean | 24.17 / 521.23 / 180.96 | |

(*Continued*)

**Table 3.** (Continued)

| | Dataset 1 | Dataset 2 |
|---|---|---|
| | PARP2-QFRD/Nuc165 | PARP2-QFRD/Nuc165 |
| DNA B factors ($Å^2$) (min/max/mean) | 60.99 / 771.83 / 271.27 | |
| Model-to-map real-space CC (mask) | 0.61 | |
| Model-to-map real-space CC (box) | 0.79 | |
| Model-to-map real-space CC (peaks) | 0.42 | |
| Model-to-map real-space CC (volume) | 0.53 | |

residue of PARP2 (E558) is 63 Å. The fully elongated H3 N-tail would span ~ 110 Å between residues K36 and S10, the main site of histone ADP-ribosylation [19]. This indicates that, in the PARP2 –Nuc165 complex, H3S10 lies well within reach of the catalytic domain of PARP2 thanks to the structural flexibility of the H3 tail [25, 26].

## HPF1, PARP2-QFRD and Nuc165 form an enzymatically active complex in solution

Given the important role of HPF1 in promoting PARylation of chromatin (primarily on H3S10) in response to DNA damage [18, 19], we next biochemically characterized the complex formed by HPF1, PARP2-QFRD and Nuc165. HPF1 binds to the PARP2-QFRD•Nuc165 complex, as indicated by EMSA showing a well-defined slow migrating band (Fig 4A and S4A Fig). The presence of both PARP2 and HPF1 in this super-shifted complex was verified by Alexa488 (yellow) and Alexa647 (red) fluorescence, respectively. Co-elution of HPF1 with the PARP2-QFRD•Nuc165 complex in analytical size exclusion chromatography could only be detected at micromolar concentration of complex and high molar excess of HPF1, indicating weak binding (Fig 4B and S4B Fig). Analytical ultracentrifugation detected changes in sedimentation coefficient due to HPF1 binding even at 100 nM of complex, and further showed that the HPF1•PARP2-QFRD•Nuc165 ternary complex is homogeneous, as indicated by a narrow sedimentation coefficient distribution (Fig 4C). We could replicate previously reported results [19] showing that only in the presence of HPF1 is PARP2 able to ADP-ribosylate a peptide encompassing residues 1–21 of histone H3 (Fig 4D and S4C Fig). Moreover, we also observed modification in *trans* of histones in an assembled Nuc165 (Fig 4E and S4D Fig). The apparent reduction in auto-modification of PARP2 in presence of HPF1 was also observed before for PARP1 [19]. This demonstrates that HPF1, PARP2-QFRD and Nuc165 form an enzymatically active complex *in vitro*.

## Quantitative characterization of the interactions within the HPF1•PARP2-QFRD•Nuc165 complex

We next set out to quantitatively characterize the interactions of HPF1 with the components of the PARP2-QFRD•Nuc165 complex. Fluorescence polarization (FP) binding assays showed that HPF1 binds to Nuc165 with a $K_D$ of 1451.8 ± 148.2 nM (Fig 5A and Table 1). No change in fluorescence polarization signal could be detected when HPF1 was titrated against Alexa488-labeled PARP2-QFRD or against the preformed PARP2-QFRD•Nuc165 complex (with

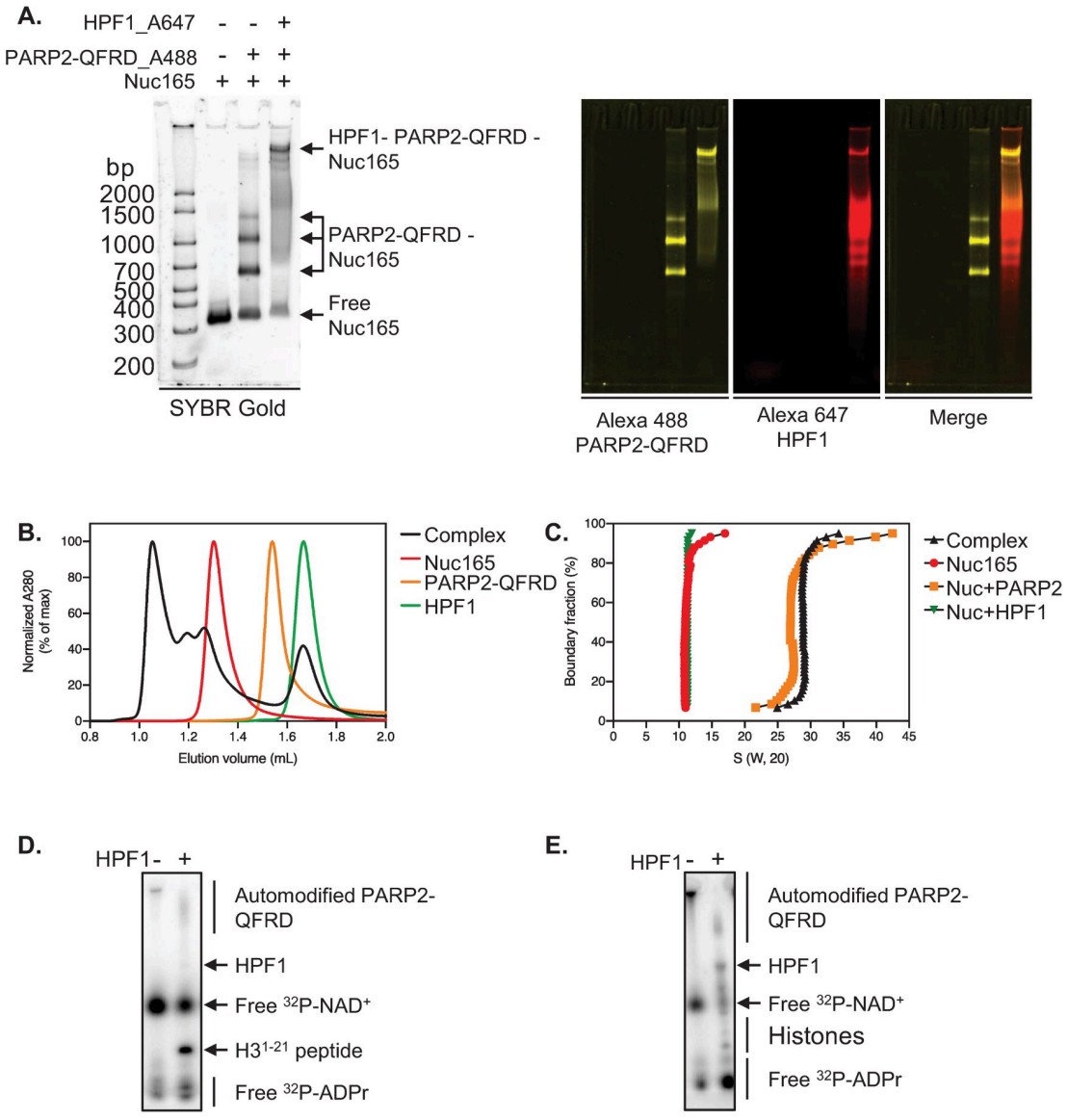

**Fig 4. HPF1, PARP2-QFRD and Nuc165 form an enzymatically active complex in solution. A:** Electrophoretic mobility shift assay of Nuc165 (1 μM), Nuc165 (1 μM) + 2 molar equivalents of PARP2-QFRD_Alexa488 (2 μM), and Nuc165 (1 μM) + 2 molar equivalents of PARP2-QFRD_Alexa488 (2 μM) + 10 molar equivalents of HPF1_Alexa647 (10 μM). The nucleosome was detected by SYBR Gold staining, PARP2-QFRD_Alexa488 and HPF1_Alxa647 were detected by their Alexa 488 and Alexa 647 fluorescence emission, respectively. **B:** Size exclusion chromatograms of HPF1 (green trace), PARP2-QFRD (orange trace), Nuc165 (red trace) and the HPF1 –PARP2-QFRD–Nuc165 complex with components mixed at molar ratios of 8:2:1 HPF1: PARP2-QFRD:Nuc165 (black trace). **C:** van Holde–Weischet plots of the sedimentation coefficient distributions of Nuc165 (red circles), Nuc165 + 16 molar equivalents of HPF1 (green inverted triangles), Nuc165 + 4 molar equivalents of PARP2-QFRD (orange squares) and Nuc165 + 4 molar equivalents of PARP2-QFRD + 16 molar equivalents of HPF1 (black triangles). The concentration of Nuc165 was 100 nM in all these experiments. **D:** Autoradiogram of an ADP-ribosylation reaction by PARP2-QFRD of a peptide of H3$^{1-21}$ (44.35 μM), in absence and presence of HPF1. p18mer DNA was used as activator. **E:** Autoradiogram of an ADP-ribosylation reaction by PARP2-QFRD of histones in Nuc165 (1.77 μM), in absence and presence of HPF1. Nuc165 was used as both activator and substrate.

Alexa488-labeled nucleosome), likely due to the already high fluorescence polarization of these probes in absence of HPF1 rather than a lack of binding, since we already established binding (Fig 4).

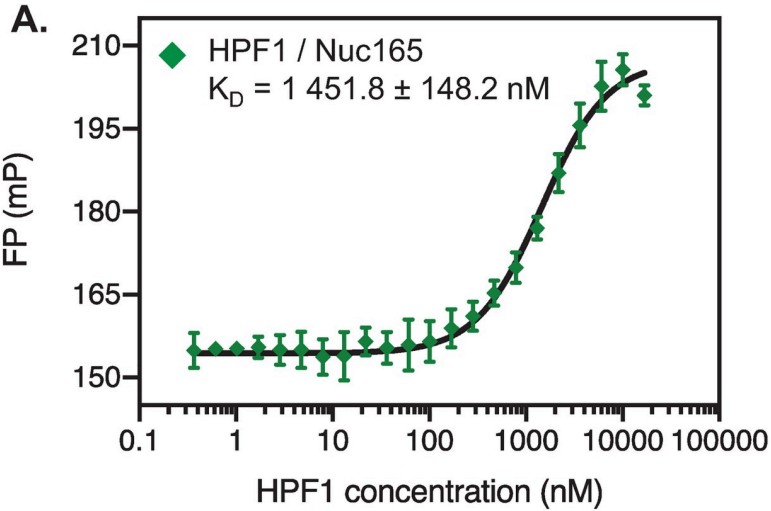

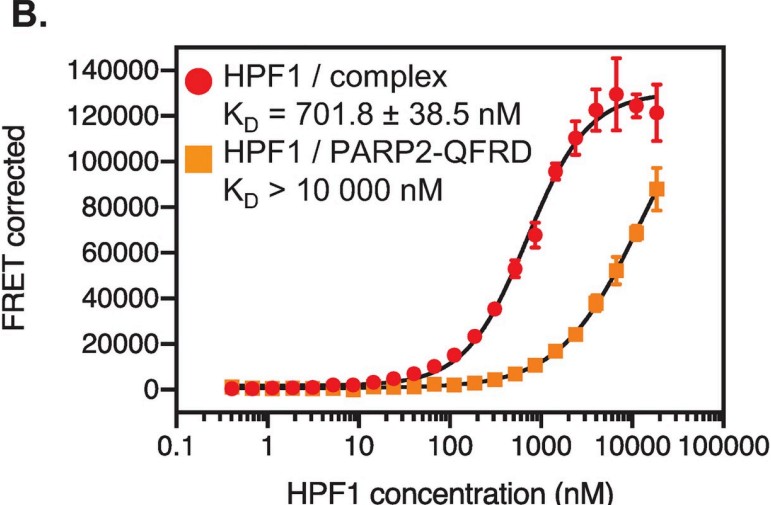

**Fig 5. Quantitation of the interactions within the HPF1 –PARP2-QFRD–Nuc165 ternary complex. A:** Fluorescence polarization binding curve of HPF1 to Alexa488-labeled Nuc165 (5 nM). **B:** FRET binding curves of HPF1_A647 to the preformed PARP2-QFRD_A488 –Nuc165 complex (1000 nM PARP2-QFRD, 100 nM Nuc165) and PARP2-QFRD_A488 (1000 nM). Points and error bars in both panels are the mean and standard deviation from three independent measurements (no visible error bar means that the error bar is smaller than the symbol used to plot the data point). Reported $K_D$ values are the mean and standard error of the mean. All $K_D$ values are listed in Table 3.

To work around this lack of FP signal, we established a binding assay of HPF1 to the pre-formed PARP2-QFRD•Nuc165 complex based on fluorescence resonance energy transfer (FRET). We used Alexa488-labeled PARP2-QFRD as the donor and Alexa647-labeled HPF1 as the acceptor. HPF1 has a very weak affinity for PARP2-QFRD in absence of Nuc165 ($K_D >$ 10 μM) but binds to the pre-formed PARP2-QFRD•Nuc165 complex with a $K_D$ of 701.8 ± 38.5 nM (Fig 5B and Table 1). Therefore, the damaged chromatin-bound conformation of PARP2-QFRD causes a more than 14-fold enhancement in the affinity of HPF1 for PARP2-QFRD. This finding is in agreement with the recently published structure of PARP2 bound to HPF1, a complex that could only be isolated in the presence of DNA [34]. The relatively weak binding affinities of HPF1 for PARP2, the nucleosome and the PARP2•nucleosome complex explain

why *in vitro* ADP-ribosylation reactions, which are typically performed at low PARP concentration, require a 10- to 20-fold excess of HPF1 [19, 35; this study]. Binding of HPF1 to PARP2-WT follows the same trend, with slightly higher affinities ($K_D > 3$ μM for PARP2-WT, $K_D = 278.9 \pm 16.6$ nM for the pre-formed PARP2-WT•Nuc165 complex, S5 Fig and Table 1). This suggests that the Q112R-F113D mutations slightly weakened the affinity for HPF1, indicating that the WGR domain may be involved in the interaction with HPF1.

## Discussion

Here we show that PARP2 is capable of bridging DNA ends even when these ends are short (10 bp) linker arms of a nucleosome, mimicking a DSB in chromatin. This is the first example of a protein engaging in a DNA end-joining nucleosome-binding mode, in addition to the other nucleosome-binding modes already described structurally [reviewed in 36]. Our results extend previous findings that the WGR domain of PARP2 mediates DNA end bridging of naked DNA [24], are in agreement with another study independently carried out [32], and raise the possibility that DNA bridging is a physiologically relevant feature of PARP2 in the context of chromatin. DNA end-joining is indeed a widespread property among DSB repair factors: Mre11 promotes DNA synapsis [37], Rad50 promotes long- and short-range tethering of DNA ends [38, 39], CtIP directs DSB repair to the homology-directed repair pathway through DNA bridging [40, 41], and most proteins of the non-homologous end-joining pathway exhibit DNA tethering activities [42, 43]. Given that PARP2 recruits to DNA breaks faster than these factors [reviewed in 4, see also 44], within seconds of the occurrence of DNA damage, we speculate that it is the factor most likely to tether the ends of a DSB. However, PARP2 critically needs a terminal 5'-phosphate group to bind DNA with maximal affinity [11] and bridges blunt ends [24], which suggests that it would be unable to efficiently bridge breaks with complex ends (dephosphorylated, with single-strand overhangs, or with bulky adducts). With such complex DNA ends, proper repair would rely on downstream factors with both end-bridging and end-processing activities, such as the Mre11-Rad50 complex. To test the physiological relevance of DNA end-bridging by PARP2 *in vivo*, one would need to assess the survival after DNA damage of cells expressing a mutant of PARP2 unable to bridge DNA ends, but still be enzymatically active to ensure that any observed phenotype does not arise from a lack of PARylation. All point mutations described so far that disrupt DNA bridging also disrupt catalytic activity [24], and separating these two functions may not be possible given the functional coupling between the WGR and catalytic domains [11, 45].

HPF1 binds to nucleosomes and to PARP2 rather weakly ($K_D = 1451.8 \pm 148.2$ nM and $K_D > 3$ μM, respectively), but it binds to the preformed PARP2•nucleosome complex with a much higher affinity of $K_D = 278.9 \pm 16.6$ nM. This 10-fold enhancement in affinity indicates a preference of HPF1 for the DSB-bound conformation of PARP2, consistent with its importance to elicit a proper DNA damage response [18, 19]. Although HPF1, PARP2-QFRD and Nuc165 form a homogeneous complex, we could not detect HPF1 in any of our cryo-EM maps. HPF1 likely dissociates during blotting, due to its relatively weak binding affinity for the PARP2-QFRD–Nuc165 complex. Structural information on HPF1 and how it interacts with PARP2 and the nucleosome will be key to understand the molecular mechanism of chromatin ADP-ribosylation [32].

Based on previous literature and our study, we propose a model of the sequence of events in the early steps of the DNA damage response. After occurrence of a DSB, PARP1 recruits to the site within seconds. It is likely the first factor to recruit because it is the most abundant nuclear protein after histones, diffuses fast and has a high affinity for DNA breaks [4, 8, 46]. The DSB-bound conformation of PARP1 releases its auto-inhibition and initiates self-PARylation [21,

45]. HPF1 recruits to the site soon after PARP1 [8, 18] and shifts its substrate specificity towards histones [19]. PARylated PARP1 eventually dissociates from the DSB [7, 11], while PARP2 recruits to the site at a slower rate compared to PARP1 [7, 8] by binding to the PAR chains initially deposited by PARP1 [9]. PARP2 then binds to and tethers the ends of the DSB, consistent with the observation that PARP2 persists longer than PARP1 at DNA damage sites in cells [7]. The conformation of PARP2 bound to damaged chromatin releases its auto-inhibition [11, 45] and also enhances its binding affinity for HPF1 by at least 10-fold, enabling it to PARylate histones [19]. PARP2 also builds branched PAR chains on the PAR initially deposited by PARP1 [9]. PARylated PARP2 eventually dissociates [11], leaving the break site surrounded by PARylated chromatin competent for the recruitment of downstream repair factors. Such factors possess PAR-binding domains and include the PAR-binding Zn-finger-containing nuclease APLF [47–50], the macro domain-containing chromatin remodeler ALC1 [51–56] and many others.

## Materials and methods

### Cloning and mutagenesis

The pET28b bacterial expression vector containing the cDNA of human PARP2 (isoform 2) was a kind gift from Dr John Pascal (University of Montréal). The PARP2$^{Q112R, F113D}$ double point mutant was generated from this vector using QuikChange Mutagenesis (Agilent) following the manufacturer's instructions. A vector containing a fusion construct of dsRed with human HPF1 was kindly provided by Dr Shan Zha (Columbia University). The cDNA of HPF1 was amplified by PCR from this plasmid and introduced by restriction cloning between the *Nde1* and *Xho1* sites into the pET28a bacterial expression vector. All constructs were verified by Sanger DNA sequencing.

### Protein expression and purification

PARP2-WT, PARP2-QFRD, and HPF1 were expressed in *E. coli* Rosetta2 (DE3) pLysS grown at 37˚C to an $OD_{600 nm}$ of ~1, with expression induced by addition of IPTG to 0.5 mM (final) and further grown for 3 hours at 37˚C (for PARP2) or overnight at 18˚C (for HPF1) before harvesting. PARP2-WT and PARP2-QFRD were purified following the same procedure, all steps were performed in a cold room or on ice. Bacterial pellets were resuspended in lysis buffer (Tris 50 mM, pH = 7.5, NaCl 500 mM, imidazole 60 mM, TCEP 0.5 mM, AEBSF 1 mM), supplemented with one tablet of protease inhibitor cocktail (cOmplete EDTA-free, Roche) per pellet, NP-40 to 0.1% v/v, MgCl$_2$ to 1.5 mM final and 1 µL of benzonase nuclease per pellet. Lysis was performed by sonication with a Branson Digital Sonifier device used at 60% power with pulses of 1 s interspersed with pauses of 1 s, until the suspension was not viscous anymore (typically between 5 min and 10 min of total pulse time). Crude lysates were clarified by centrifugation at 18,000 rpm on a JA-20 rotor (Beckman-Coulter) for 30 min at 4˚C. Clear lysates were filtered through 0.45 µm and immediately applied at 2 mL/min onto a HisTrap HP 5 mL column (GE Healthcare) previously equilibrated in lysis buffer. Unbound contaminants were washed by flowing lysis buffer until absorbance at 280 and 260 nm returned to a flat baseline, then elution was performed by a gradient to 1 M imidazole over 20 column volumes. Fractions of interest were identified by SDS-PAGE, pooled, diluted by half to lower NaCl concentration to 250 mM and applied at 2 mL/min onto a HiTrap Heparin HP 5 mL column (GE Healthcare) previously equilibrated in Tris 50 mM, pH = 7.5, NaCl 250 mM, EDTA 0.1 mM, TCEP 0.1 mM, AEBSF 0.1 mM. Unbound contaminants were washed by flowing Heparin buffer until absorbance at 280 and 260 nm returned to a flat baseline, then elution was performed by a gradient to 1 M NaCl over 20 column volumes. Fractions of interest were

identified by SDS-PAGE, pooled, concentrated using centrifugal filters to bring down the volume to ~1 mL and injected onto a Superdex 200 16/60 size exclusion column (GE Healthcare) previously equilibrated in HEPES 50 mM, pH = 7.5, NaCl 150 mM, EDTA 0.1 mM, TCEP 0.1 mM, AEBSF 0.1 mM. Fractions of interest were identified by SDS-PAGE and pooled. Protein concentration was determined by absorbance at 280 nm. Purified proteins were aliquoted, flash frozen in liquid nitrogen and stored at -80°C.

HPF1 was purified the same way, with the following differences: all buffers were adjusted to a pH of 8 instead of 7.5, the Heparin affinity step was skipped, a Superdex 75 size exclusion column was used instead of a Superdex 200, and the size exclusion chromatography buffer contained Tris at pH = 8 instead of HEPES at pH = 7.5, to keep the pH more than 1 unit away from the pI of HPF1 (6.77).

Absence of DNA contamination in purified proteins was verified by UV absorbance: the A260/A280 ratio was typically between 0.49 and 0.55 for all purified proteins. Purity of the purified proteins was assessed by SDS-PAGE (S7A and S7B Fig).

### Fluorescent labeling of recombinant proteins

PARP2 was mixed at equimolar ratio with Alexa Fluor 488 C5 maleimide. HPF1 was mixed at equimolar ratio with Alexa Fluor 647 C2 maleimide. These mixtures were incubated one hour at 4°C on a rotating wheel. Remaining free dye was removed by several rounds of dilution in storage buffer and concentration on a centrifugal filter (Amicon Ultra-4, Millipore; 30 kDa MWCO for PARP2, 10 kDa MWCO for HPF1) until no dye absorbance was detectable in the concentration flow through. The extent of labeling, defined as the ratio of the molar concentration of dye versus protein, was determined by UV-visible spectrophotometry and was typically between 20 and 80%. Covalent labeling was verified by SDS-PAGE (S7A and S7B Fig), and fluorescently labeled proteins were shown to be enzymatically active (S8 Fig).

### Nucleosome reconstitution

Nucleosomes were assembled using the continuous salt gradient dialysis method described previously [57], from recombinant human histone octamer and a 165 bp long DNA harboring the 601 nucleosome positioning sequence [58]. This sequence positions the histone octamer in a way that leaves two linker arms of 7 and 11 bp (the entire sequence is shown in Supplementary information). We call this nucleosome Nuc165. Purity of salt-reconstituted nucleosomes was assessed by native PAGE (S7C Fig). Nuc165 labeled with Alexa488 was assembled the same way, using a human histone octamer previously labeled on H2B-T112C with Alexa Fluor 488 C5 maleimide (about 50% labeling efficiency). Covalent labeling was verified by native PAGE (S7D Fig).

### EMSA

PARP2 –Nuc165 complexes were prepared in TCS buffer (Tris 20 mM, pH = 7.5, EDTA 1 mM, DTT 1 mM) in a final volume of 5 μL, at a final nucleosome concentration of 1.4 μM with 0.5, 1, 1.5 and 2 molar equivalents of PARP2-WT or PARP2-QFRD. Complexes were incubated at 4°C for 30 min before separation by electrophoresis on 5% native PAGE at 150 V constant in TBE 0.2X buffer at 4°C. Complexes were detected by ethidium bromide staining.

HPF1_A647 –PARP2-QFRD_A488 –Nuc165 complexes were prepared in TCS buffer (Tris 20 mM, pH = 7.5, EDTA 1 mM, DTT 1 mM) in a final volume of 5 μL, at a final nucleosome concentration of 1 μM with 2 molar equivalents of PARP2-QFRD_A488 and 10 molar equivalents of HPF1_A647. Complexes were incubated at 4°C for 60 min before separation by electrophoresis on 5% native PAGE at 150 V constant in TBE 0.2X buffer at 4°C. Complexes were

detected by SYBR Gold staining and fluorescence detection of the Alexa488 and Alexa647 labels.

## Thermal denaturation assay

PARP2-WT and PARP2-QFRD were diluted to 0.2 mg/mL with storage buffer (HEPES 50 mM, pH = 7.5, NaCl 150 mM, EDTA 0.1 mM, TCEP 0.1 mM). 12.5 μL of these protein solutions were mixed with 5 μL of thermal shift assay buffer and 2.5 μL of 16X thermal shift assay dye, following the manufacturer's protocol (Protein Thermal Shift Dye Kit, Applied Biosystems), in four technical replicates. Fluorescence was monitored as a function of increasing temperature in a qPCR instrument. $T_m$ values were determined as the temperature at which the derivative of the melting curve peaks. Reported $T_m$ values are the mean and standard deviation of values obtained from three independent measurements performed on different days.

## Conservation analysis

Conservation analysis was performed with the ConSurf web server [59], using default parameters and extracting the sequence of chain A from PDB entry 6F5B to perform the initial BLAST search. Identifiers and residue ranges of all protein sequences used in this analysis are listed in Supplementary Information. S1L Fig of PDB entry 6F5B were made with PyMOL version 2.3.3 (Schrodinger LLC).

## PARP2 activity as detected by the smear assay

PARP2-WT or PARP2-QFRD was mixed with NAD$^+$ and p18mer DNA at final concentrations of 1 μM PARP2, 1 μM p18mer and 500 μM NAD$^+$ in 50 mM Tris-HCl, pH = 8, 100 mM NaCl, 1 mM MgCl$_2$, in a total volume of 20 μL. Reactions were incubated 2 hours at room temperature and arrested by addition of Laemmli buffer and boiling for 5 min. Reaction products were separated by SDS-PAGE and proteins were detected by Coomassie staining or fluorescence detection of the Alexa488 label.

## PARP2 activity as detected by incorporation of $^{32}$P-ADPr

PARP2 (30 nM final) was pre-incubated with varying concentrations of p18mer DNA (0.01–200 nM final; sequence of p18mer: top strand 5′ P-GGGTTGCGGCCGCTTGGG-3′ OH, bottom strand 5′ -CCCAAGCGGCCGCAACCC-3′ OH, purchased as a duplex from Integrated DNA Technologies) in a buffer containing 50 mM Tris pH = 8, 50 mM NaCl, 1 mM MgCl$_2$, 0.1 mM EDTA and 0.5 mg/mL bovine serum albumin in 96-well plates (Costar 3898). Following addition of $^{32}$P-NAD$^+$ (40–80 μM, 1.8 x 10$^6$ cpm/well) to yield a final volume of 25 μL, reactions were quenched after 30 s by addition of 50 μL of 30% trichloracetic acid (TCA). Samples (50 μL of total) were then loaded onto a Whatman Mini-Fold Spot-Blot apparatus containing a Whatman GF/C glass microfiber filter. Each well was washed three times with 100 μL of 10% TCA. After removal of the filter from the apparatus, the filter was gently incubated in 20–40 mL of 10% TCA for three more washes. After drying, the filter was exposed to a phosphoimager screen (GE Healthcare) and imaged using a Typhoon 9500. Spot intensities were quantitated using ImageQuant.

## Fluorescence polarization binding assays

All fluorescence polarization (FP) binding assays were performed in 50 mM Tris-HCl, pH = 8, 50 mM NaCl (except for salt concentration series experiments, as noted), 1 mM MgCl$_2$, 0.1 mM EDTA, 0.1 mM TCEP and 0.01% v/v NP-40.

For DNA binding experiments, PARP2-WT and PARP2-QFRD were serially diluted in rows of a 384-well plate (Corning, product reference 3575) by a factor of 15/25 from 2000 nM down to 0 nM, in a final volume of 10 μL, and mixed with 10 μL of fluorescein-labeled p18mer DNA at 6 nM (sequence of p18mer_fluorescein: top strand 5′ P-GGGTTGCGGCCGCTTGGG-3′ OH, bottom strand 5′-6FAM-CCCAAGCGGCCGCAACCC-3′ OH, purchased as a duplex from Integrated DNA Technologies). The resulting final concentration series of protein ranged from 1000 nM to 0 nM and the resulting final concentration of labeled p18mer DNA probe was 3 nM.

Nucleosome binding assays were prepared the same way, with the same concentration series of PARP2-WT and PARP2-QFRD. Alexa488-labeled Nuc165 was used as probe at a final concentration in the assay of 5 nM.

All titrations of HPF1 were prepared the same way (serial dilution by a factor of 15/25), with a final probe concentration of 5 nM (fluorescein-labeled p18mer or Alexa488-labeled Nuc165) and with HPF1 ranging from 16.8 μM to 0 μM final concentration in the assay.

For each experiment, fluorescence polarization (FP) was measured for two technical replicates of the titration series in a BMG Labtech CLARIOstar plate reader, with final FP values taken as the average of these two technical replicates. The fluorescence polarization baseline was arbitrarily set to 70 mP for fluorescein-labeled p18mer. The fluorescence polarization of Alexa488-labeled Nuc165 was measured relative to that of fluorescein-labeled p18mer, and a value of 157 mP was used as baseline for nucleosome binding experiments. Each experiment was repeated three times from independent dilutions series prepared on different days.

## FRET binding assays

All fluorescence resonance energy transfer (FRET) binding assays were performed in 50 mM Tris-HCl, pH = 8, 50 mM NaCl, 1 mM MgCl$_2$, 0.1 mM EDTA, 0.1 mM TCEP and 0.01% v/v NP-40.

HPF1 was serially diluted in rows of a 384-well plate (Corning, product reference 3575) by a factor of 15/25 from 37 μM down to 0 μM, in a final volume of 10 μL, and mixed with the preformed PARP2-QFRD–Nuc165 complex (2 μM / 200 nM, giving full saturation of the nucleosome by PARP2) or with PARP2-QFRD alone (2 μM). The resulting final concentration series of HPF1 ranged from 18.5 μM to 0 μM, the resulting final concentration of PARP2-QFRD–Nuc165 complex was 1 μM / 100 nM, and the resulting final concentration of PARP2-QFRD was 1 μM. Each experiment consisted of three rows: the first row (donor + acceptor) contained Alexa488-labeled PARP2-QFRD (donor) and Alexa647-labeled HPF1 (acceptor), the second row (donor + unlabeled) contained Alexa488-labeled PARP2-QFRD (donor) and unlabeled HPF1, and the third row (unlabeled + acceptor) contained unlabeled PARP2-QFRD and Alexa647-labeled HPF1 (acceptor).

Fluorescence intensity was recorded using a BMG Labtech CLARIOstar plate reader at three distinct settings of excitation and emission (noted as excitation wavelength–bandwidth / dichroic wavelength / emission wavelength–bandwidth, all in nm), in this order: 488–20 / 530 / 680–50 (FRET channel), 620–30 / 645 / 680–50 (acceptor channel) and 488–20 / 509 / 535–30 (donor channel). The gain was reset for each experiment, using the well with the maximum acceptor concentration in the donor + acceptor row to set the gain of the FRET channel, using the well with the maximum acceptor concentration in the acceptor + unlabeled row to set the gain of the acceptor channel, and using a well with no unlabeled HPF1 in the donor + unlabeled row to set the gain of the donor channel.

Raw fluorescence intensity values were used to correct the raw FRET signal for the donor bleed-through and for the acceptor direct excitation, as described previously [60]. The binding

curve corresponds to the resulting corrected FRET values plotted against the concentration series of Alexa647-labeled HPF1. Each experiment was repeated three times from independent dilutions series prepared on different days.

## Fitting of binding curves

We first attempted to fit the following quadratic model to the experimental FP and FRET binding curves:

$$S = S_{min} + (S_{max} - S_{min}) \times \frac{(K_D + R_{tot} + L_{tot}) - \sqrt{(-K_D - R_{tot} - L_{tot})^2 - 4 \times R_{tot} \times L_{tot}}}{2 \times R_{tot}} \quad \text{Eq 1}$$

In which $S$ denotes the experimental signal (fluorescence polarization or corrected FRET; Y axis), $S_{min}$ and $S_{max}$ denote respectively the minimum and maximum plateaus of the binding curve, $K_D$ denotes the equilibrium dissociation constant, $R_{tot}$ denotes the constant total concentration of receptor (for FP binding assays, this is the concentration of fluorescein-labeled p18mer DNA or Alexa488-labeled Nuc165 probe; for FRET binding assays, this is the concentration of Alexa488-labeled PARP2-QFRD or PARP2-QFRD/Nuc165 complex) and $L_{tot}$ denotes the total concentration of ligand at each titration point (here, the concentration of protein across the titration series; X axis).

However, some of our binding curves had a slope with a pronounced systematic deviation from the quadratic model (S1D, S1G and S1H Fig), which justified fitting the following Hill model instead:

$$S = S_{min} + (S_{max} - S_{min}) \times \frac{L^h}{K_D^{\,h} + L^h} \quad \text{Eq 2}$$

In which $L$ denotes the concentration of free ligand at equilibrium at each titration point (in this case, approximated by the total concentration of protein; X axis), $h$ denotes the Hill coefficient and all other symbols are as defined in the previous equation.

For consistency, we used the Hill model to fit all binding curves and derive all $K_D$ values presented in this study. The Hill coefficients close to 2 obtained for the PARP2 –Nuc165 interaction are in agreement with the 2:2 stoichiometry observed by SEC-MALS and cryo-EM.

## SEC-MALS

The following samples were prepared in a final volume of 110 μL: PARP2-QFRD at 30 μM (2 mg/mL), Nuc165 at 4.61 μM (0.97 mg/mL), Nuc165 at 4.5 μM + 0.5 molar equivalent PARP2-QFRD, Nuc165 at 4.5 μM + 1 molar equivalent PARP2-QFRD, Nuc165 at 4.4 μM + 2 molar equivalents PARP2-QFRD and dinucleosome 2x165 at 2.4 μM (1.09 mg/mL). Mixtures of Nuc165 and PARP2-QFRDat micromolar concentrations typically turned cloudy upon mixing and were allowed to clear overnight at 4˚C. All samples were centrifuged 5 min at 16 000 g at 4˚C before injection.

Samples were injected at 0.75 mL/min on a Superdex 200 Increase 10/300 size exclusion column equilibrated in HEPES 50 mM, pH = 7.5, NaCl 150 mM, MgCl₂ 1 mM, EDTA 0.1 mM, TCEP 0.1 mM. Light scattering and differential refractive index were continuously measured downstream of the column with a Dawn Heleos-II instrument (Wyatt Technology) and an Optilab rEX instrument (Wyatt Technology), respectively. Light scattering and differential refractive index signals were processed and analyzed in Astra version 7 software (Wyatt Technology). Bovine serum albumin (BSA) at 2 mg/mL was injected before the first sample and after the last sample to validate instrument performance and normalize signals. Experimental

molecular weight determination for the BSA monomer and dimer typically yielded values within 3% of theoretical values predicted from the sequence. The discrepancy between the experimentally determined MW value ($MW_{exp}$) and the MW value calculated from sequences and stoichiometries of components ($MW_{th}$) was calculated as:

$$discrepancy = 100 \times \frac{MW_{exp} - MW_{th}}{MW_{th}}$$

Stoichiometries reported in Table 1 are the ones that minimize the discrepancy between experimental and theoretical molecular weight.

## Analytical size exclusion chromatography

20 μL of proteins, nucleosomes and complexes were injected in a Superdex 200 Increase 3.2/300 size exclusion column (GE Healthcare) at 0.075 mL/min, while collecting fractions of 100 uL across the entire chromatogram. HPF1, PARP2-QFRD and Nuc165 were individually run in Tris 50 mM, pH = 8, NaCl 150 mM, MgCl$_2$ 1 mM, EDTA 0.1 mM, TCEP 0.1 mM. The HPF1 –PARP2-QFRD–Nuc165 complex was run in the same buffer but with only 50 mM NaCl. HPF1 was injected at an initial concentration of 16.8 μM, PARP2-QFRD at an initial concentration of 17.9 μM and Nuc165 at an initial concentration of 2.7 μM. The HPF1 – PARP2-QFRD–Nuc165 complex was prepared at initial concentrations of each component of 18.98, 4.75 and 2.37 μM, respectively (i.e. at molar ratios of 8:2:1) in a final volume of 200 μL. The complex was then concentrated 10-fold down to the volume of one injection using a 10 kDa cutoff centrifugal filter.

## Analytical ultracentrifugation

Alexa488-labeled Nuc165 was incubated with 4 molar equivalents of PARP2-QFRD in a buffer containing 50 mM HEPES 7.5, 150 mM NaCl, 1 mM MgCl$_2$, 0.1 mM TCEP and 0.075 mg/mL BSA. The final concentration of nucleosome was 100 nM, and because their labeling efficiency was 50% the final concentration of label was 50 nM. This complex was further incubated with increasing amounts of HPF1 (1, 2, 4, 8 or 16 molar equivalents over PARP2-QFRD). Additionally, Alexa488-labeled Nuc165 was incubated with 16 molar equivalents of HPF1 in the absence of PAPR2-QFRD to test for a direct interaction between HPF1 and the nucleosome. The samples were incubated at room temperature for 30 minutes and loaded into AUC cells. To confirm formation of complexes, an aliquot of each sample was also run on a 5% native PAGE in 0.2X TBE at 150 V for 70 minutes, and the gel was subjected to Alexa488 fluorescence detection, was then stained with SYBR Gold for DNA detection, and was finally stained with Blazin Blue for protein detection.

The samples were spun at 25 000 RPM at 20˚C in an An50Ti rotor in a Beckman XLA analytical ultracentrifuge fitted with an Aviv fluorescence detector. Fluorescence scans were collected at 0.003 cm intervals until solutes were completely sedimented. Data were processed using Ultrascan III [61]. Sedimentation coefficients were obtained with enhanced van Holde–Weischet analysis [62] and graphs were plotted in Graphpad Prism.

## HPF1 activity assays

ADP-ribosylation reactions of the H3$^{1-21}$ peptide contained 100 nM PARP2-WT or PARP2-QFRD, 100 nM p18mer DNA as activator, 44.35 μM (0.1 mg/mL) of H3$^{1-21}$ peptide as substrate (purchased from Eurogentec), no HPF1 or 2 μM HPF1, and 20 μM NAD$^+$ in a final volume of 20 μL. ADP-ribosylation reactions of Nuc165 contained 100 nM PARP2-WT or

PARP2-QFRD, 5.66 μM Nuc165 as both activator and substrate, no HPF1 or 3.13 μM HPF1, and 50 μM NAD$^+$ in a final volume of 20 μL. NAD$^+$ was added last to initiate the reaction, as a mixture of $^{32}$P-NAD$^+$ and NAD$^+$ at a 1:100 molar ratio. Reactions were incubated at room temperature for 75 minutes, then 10 μL of each reaction were quenched by addition of Olaparib to a final concentration of 2 μM, boiled with Laemmli buffer (1X final) for 5 min, and half of the resulting denatured sample was run on SDS-PAGE (4–12% Bis-Tris SDS gel in MES buffer) at 200 V for 30 min. Gels were imaged with a phosphorescent screen in a Typhoon scanner (GE Healthcare) with PMT gain set to 1000.

## Cryo-EM sample preparation and screening

PARP2-QFRD and Nuc165 were mixed at final concentrations of 6.3 and 3.15 μM, respectively (a 2:1 molar ratio), in TCS buffer (Tris 20 mM, pH = 7.5, EDTA 1 mM, DTT 1 mM) supplemented with 1 mM MgCl$_2$. The salt present in the protein storage buffer brought the final NaCl concentration of this cryo-EM sample to 5.3 mM. 4 μL of this complex were applied on freshly glow-discharged Quantifoil R 1.2/1.3 300 Mesh Copper holey carbon grids (previously cleaned with chloroform and allowed to dry overnight) and vitrified using a Vitrobot Mark IV instrument with the chamber equilibrated at 95% relative humidity and 4°C, using a blot time of 2 s and a blot force of 0.

Grids were screened on a Tecnai F20 microscope (field emission gun, 200 kV acceleration voltage) equipped with a Gatan US4000 4k x 4k CCD camera, using a magnification of 62 000x, a defocus of -3 μm and a total dose of 70 electrons per square angstrom.

## Cryo-EM data collection and structure determination

For PARP2-QFRD–Nuc165 (dataset 1), data collection was performed on a Tecnai F30 microscope (field emission gun, 300 kV acceleration voltage) equipped with a Gatan K2 Summit direct electron detector operated in counting mode, using a magnification of 31,000x (resulting in a pixel size of 1.271 Å/pixel), a nominal defocus range of -1 to -2.5 μm, recording 40 movie frames per field of view with a dose of 1.4025 electrons per square angstrom per frame (total dose of 56.1 electrons per square angstrom). A representative micrograph is shown in S3A Fig (after motion correction) and its power spectrum and CTF estimation is shown in S3B Fig.

Movie frames were aligned using MotionCor2 [63] and CTF estimation was performed using Gctf [64], all from within RELION version 2.1.0 [65]. All subsequent steps were performed in RELION. About 1000 single nucleosomes were picked manually and subjected to 2D classification. Among resulting 2D class averages, those displaying high-resolution features were used as templates for autopicking. The resulting set of particles was subjected to several rounds of 2D classification to screen out erroneously picked particles (frost contaminants, carbon edge, etc.). We then generated a map of a single nucleosome low-pass filtered to 50 Å resolution from PDB entry 1KX5 [66] using EMAN2's pdb2mrc.py program [67]. This map was used as a reference for 3D classification, after which we noticed additional density extending from the DNA end of the reconstructed nucleosome, suggesting the presence of another nucleosome at lower contour levels; a second nucleosome was indeed present, but clipped in half (3D classes 1 and 2 in S3F Fig).

We repeated manual and automatic particle picking, this time centering particle coordinates on nucleosome dimers. Subsequent 2D and 3D classifications revealed the entirety of the second nucleosome in the bridged nucleosome dimer particle (S3G Fig). The entire dataset was finally reprocessed, using crYOLO version 1.3.1 [68] to pick particles and RELION version 3.0 for subsequent processing (S3H Fig). The 3D reconstruction of the nucleosome dimer

bridged by PARP2-QFRD was refined to an overall resolution of 10.5 Å (S3D Fig). Post-processing and local resolution calculation were performed in RELION.

Another dataset (dataset 2) was collected independently, from a grid prepared with a different batch of complex on a different day, on a Titan Krios microscope located at the HHMI Janelia Research Campus (field emission gun, 300 kV acceleration voltage) equipped with a Gatan K2 Summit direct electron detector operated in counting mode and super-resolution, using a magnification of 22,500x (resulting in a pixel size of 1.31 Å/pixel after binning super-resolution images by a factor of 2), a nominal defocus range of -1 to -2.5 μm, recording 50 movie frames per field of view with a dose of 1.16 electrons per square angstrom per frame (total dose of 58 electrons per square angstrom). Movie frames were aligned using MotionCor2 [63] and CTF estimation was performed using Gctf [64], all from within RELION version 3.0. Particle picking was performed with crYOLO version 1.3.6 [68]. We tried using an *ab initio* reconstruction and PDB entry 1KX5 low-pass filtered to 30 Å as a 3D reference for 3D classification, but due to a large proportion of damaged particles in this dataset, 3D classification could only converge when we used the nucleosome dimer map obtained from dataset 1 (low-pass filtered to 30 Å) as a 3D reference. The final 3D reconstruction from dataset 2 was refined to an overall resolution of 10.5 Å, with local resolution in a similar range as the reconstruction from dataset 1 (S3K Fig). These two reconstructions superimpose with a real-space correlation coefficient of 0.973 at a contour level of 0.005 for both maps, as calculated with UCSF Chimera version 1.13.1.

Figures displaying maps and models were produced with UCSF Chimera, version 1.13.1 [69]. Histograms of local resolution values were computed using the Python libraries *mrcfile* [70], *NumPy* and *matplotlib* [71], using a script provided by Twitter user @biochem_fan (also in Supplementary Information).

## Model building and refinement

We built a model of the human 601 nucleosome using the 147 bp Widom 601 DNA from PDB entry 6R1T [72] and the human histone octamer from PDB entry 5Y0C [73]. Nucleosomal DNA was oriented to fit the human histone octamer by aligning core histone-fold backbone positions (resulting in a backbone RMSD of 0.8 Å).

We then performed rigid-body fitting of two copies of this nucleosome model and of the PARP2 WGR domain in complex with DNA (PDB entry 6F5B) [24] in our cryo-EM map, using UCSF Chimera version 1.13.1 [69]. Inspection of the region of linker DNA where PARP2 connects the two nucleosomes indicated that a linker arm of 8 bp best explains the map, while Nuc165 has linker arms of 7 and 11 bp. This reveals that the map is an average of potentially three configurations of the nucleosome dimer: a dimer bridged either by two long linkers, by two short linkers, or by one long and one short linker. The best-fitting linker length of 8 bp is shorter than the average between 7 and 11 bp (9 bp), indicating a preference of PARP2 to bridge short linker ends (because the map is an average weighted by the number of particles in each possible configuration of the dimer). This intrinsic heterogeneity limits the resolution of our map, as well as missing orientations due to the small number of particles in the final 3D class (S3M Fig). We decided to model linker arms of 8 bp because this length best fits the map without imposing significant strain to the DNA backbone conformation.

The DNA ends not bound by PARP2 were modeled with linear B-DNA built using the *nucleic acid builder* program of the Amber suite version 18 [74, 75]. The DNA ends bound by PARP2 were modeled using DNA chains from PDB entry 6F5B, truncated 8 bp away from the PARP2-bound extremity. These DNA chains were mutated to match the sequence of our 165 bp 601 DNA using UCSF Chimera version 1.13.1 [69] and renumbered and assigned correct

chain identifiers to link them to the nucleosomal DNA using Coot 0.9-pre [76]. The Q112R and F113D mutations were performed in the PARP2 WGR model using Chimera.

To optimize the geometry of the linker DNA sections prior to real-space refinement, the system was modeled in a 100 mM ionic strength implicit solvent environment (igb = 5 with mbondi2 radii parameters) [77] and energy minimized with the *pmemd* program of the Amber suite version 18 [75]. Protein parameters were taken from the ff14SB force field [78] and DNA parameters were from the bsc1 force field [79]. Since terminal 5'-phosphate groups of DNA are not currently parameterized in the Amber force field, and our primary goal at this step was to correct the bridging linker DNA configurations, we opted to remove the terminal 5'-phosphate prior to energy minimization. No restraints were placed on any atomic positions during the minimization, and an infinite residue-pair cutoff was employed. Minimization was conducted for 5000 iterations, and appreciable improvements in both overall system and internal bond geometry energies were observed (S2 Table). The previously removed terminal 5'-phosphate groups were then reintroduced into the system, according to their known geometry from the crystal structure (PDB entry 6F5B). This model was used for real-space refinement in *phenix.real_space_refine* [80] from the Phenix suite version 1.17–3644. The refinement strategy included global rigid body refinement, rigid body refinement with one rigid body per nucleosome and per copy of the WGR domain, and B-factors. The refinement was performed with secondary structure, base pair and base stacking restraints. Refinement statistics are listed in Table 2.

On the basis of our refined model, we extrapolated a plausible spatial location of the catalytic domain of PARP2-QFRD by superimposing the WGR domain of PARP1 from the crystal structure of PARP1 bound to DNA (PDB entry 4DQY) [21] to the WGR domain of PARP2-QFRD in our model (RMSD = 0.92 Å between 78 C$\alpha$ pairs), then by superimposing the catalytic domain of PARP2 (PDB entry 4ZZX) [33] to the catalytic domain of PARP1 (RMSD = 0.82 Å between 263 C$\alpha$ pairs). We finally removed the structure of PARP1 from the model.

Electron microscopy was done at the University of Colorado, Boulder EM Services Core Facility in the department of Molecular, Cellular and Developmental Biology, and at the HHMI Janelia Cryo-EM Facility, with the technical assistance of facility staff.

## Supporting information

**S1 Fig. PARP2-QFRD behaves like PARP2-WT *in vitro*. A:** Cryo-electron micrograph of the PARP2-WT•Nuc165 complex. Enlarged views of two aggregates are shown (1 and 2). Scale bar: 50 nm. **B:** Thermal denaturation curves of PARP2-WT and PARP2-QFRD from differential scanning fluorimetry. Reported $T_m$ values are the mean and standard deviation from three independent measurements; only one replicate curve is shown for clarity. **C:** Fluorescence polarization binding curves of PARP2-WT and PARP2-QFRD to fluorescein-labeled p18mer DNA (3 nM). Points and error bars are the mean and standard deviation from three independent measurements (no visible error bar means that the error bar is smaller than the symbol used to plot the data point). Reported $K_D$ values are the mean and standard error of the mean. All $K_D$ values are listed in Table 1. **D:** Fluorescence polarization binding curves of PARP2-WT and PARP2-QFRD to Alexa488-labeled Nuc165 (5 nM). Points and error bars are the mean and standard deviation from three independent measurements (no visible error bar means that the error bar is smaller than the symbol used to plot the data point). Reported $K_D$ values are the mean and standard error of the mean. All $K_D$ values are listed in Table 1. **E:** Coomassie-stained SDS-PAGE of unmodified and PARylated PARP2-WT and PARP2-QFRD. **F:** Activation curves of PARP2-WT and PARP2-QFRD (30 nM) by increasing concentrations of

p18mer DNA. Points and error bars are the mean and standard deviation from three independent measurements (no error bar means the error bar is smaller than the symbol used to plot the data point). Reported $K_{act}$ values are the mean and standard error of the mean. **G:** Fluorescence polarization binding curves of PARP2-WT to fluorescein-labeled p18mer DNA (3 nM) at various NaCl concentrations. $K_D$ values at each NaCl concentration tested are listed in S1 Table. **H:** Fluorescence polarization binding curves of PARP2-QFRD to fluorescein-labeled p18mer DNA (3 nM) at various NaCl concentrations. $K_D$ values at each NaCl concentration tested are listed in S1 Table. **I:** Fluorescence polarization binding curves of PARP2-WT to fluorescein-labeled p18mer DNA (3 nM) in our condition (50 mM NaCl, 1 mM $MgCl_2$) and in an experiment replicating the condition reported by [11] (60 mM KCl, 8 mM $MgCl_2$). Points and error bars are the mean and standard deviation from three independent measurements (no error bar means the error bar is smaller than the symbol used to plot the data point). Reported $K_D$ values are the mean and standard error of the mean. **J:** Fluorescence polarization binding curves of PARP2-QFRD to fluorescein-labeled p18mer DNA (3 nM) in our condition (50 mM NaCl, 1 mM $MgCl_2$) and in an experiment replicating the condition reported by [11] (60 mM KCl, 8 mM $MgCl_2$). Points and error bars are the mean and standard deviation from three independent measurements (no error bar means the error bar is smaller than the symbol used to plot the data point). Reported $K_D$ values are the mean and standard error of the mean. **K:** PARP2 sequence alignment across several species. Q112, F113 are indicated with left-right arrow; WGR domain is labelled as such; Y188 residue is indicated with down arrow. **L:** Conservation analysis of the WGR domain of PARP2. The Q112 and F113 residues are no more conserved than average. Conserved residues W138, R140 and Y188 are also shown to highlight the contrast in conservation. Protein sequences used in this analysis are listed in Supplementary Information. **M:** Packing contacts in the crystal lattice of PDB entry 6F5B. DNA is drawn as grey cartoon, WGR domains are shown as surface with one shade of blue for each biological assembly, and the Q112 and F113 residues are colored in red.
(RAR)

**S2 Fig. Stoichiometry of the PARP2-QFRD•Nuc165 complex.** Size exclusion chromatograms and experimental molecular weights determined by SEC-MALS. All molecular weights are listed in Table 2. Stoichiometries consistent with the experimental molecular weights are depicted as cartoons. **A:** PARP2-QFRD. **B:** Nuc165 + 0.5 molar equivalent of PARP2-QFRD. **C:** Nuc165 + 2 molar equivalents of PARP2-QFRD.
(TIF)

**S3 Fig. Cryo-EM structure determination of the PARP2-QFRD•Nuc165 complex. A:** A representative motion-corrected, low-pass filtered micrograph from dataset 1 (scale bar 50 nm). **B:** Power spectrum and CTF estimation of the micrograph shown in panel A. **C:** Local resolution map and histogram of local resolution values of the PARP2-QFRD•Nuc165 complex 3D reconstruction from dataset 1. **D:** Overall resolution of the PARP2-QFRD•Nuc165 complex 3D reconstruction from dataset 1, as determined by the 0.143 FSC threshold (dotted horizontal line). **E:** Postprocessing Guinier plot for the above data. **F:** Euler angle distribution of the 27 889 particles that contributed to the 3D reconstruction of the PARP2-QFRD•Nuc165 complex from dataset 1. **G:** Flow chart of the initial analysis of dataset 1, picking single nucleosomes. **H:** Flow chart of the second analysis of dataset 1, picking nucleosome dimers. **I:** Flow chart of the final analysis of dataset 1, using all micrographs and picking nucleosome dimers with crYOLO using a model trained on this dataset. **J:** A representative motion-corrected, low-pass filtered micrograph from dataset 2. **K:** Power spectrum and CTF estimation of the micrograph shown in panel J. **L:** Local resolution map and histogram of local resolution values of the PARP2-QFRD•Nuc165 complex 3D reconstruction from dataset 2. **M:** Overall resolution of

the PARP2-QFRD•Nuc165 complex 3D reconstruction from dataset 2, as determined by the 0.143 FSC threshold (dotted horizontal line). **N:** Post processing Guinier plot for the above dataset. **O:** Euler angle distribution of the 16 304 particles that contributed to the 3D reconstruction of the PARP2-QFRD•Nuc165 complex from dataset 2. **P:** Flow chart of the final analysis of dataset 2, using all micrographs and picking nucleosome dimers with crYOLO using a model trained on datasets 1 and 2.
(RAR)

**S4 Fig. HPF1, PARP2 and Nuc165 form an enzymatically active complex in solution. A:** Electrophoretic mobility shift assay of Nuc165 (1 μM), Nuc165 (1 μM) + 2 molar equivalents of PARP2-WT_Alexa488 (2 μM), and Nuc165 (1 μM) + 2 molar equivalents of PARP2-WT_Alexa488 (2 μM) + 10 molar equivalents of HPF1_Alexa647 (10 μM). The nucleosome was detected by SYBR Gold staining, PARP2-WT_Alexa488 and HPF1_Alexa647 were detected by their Alexa 488 and Alexa 647 fluorescence emission, respectively (right panels). **B:** Unnormalized size exclusion chromatograms shown in Fig 4B and SDS-PAGE analysis of collected fractions. Blue traces represent absorbance at 280 nm, red traces represent absorbance at 260 nm. Elution volumes are indicated for each peak. **C:** Uncropped autoradiogram shown in Fig 4D. ADP-ribosylation reaction by PARP2-WT, PARP2-QFRD (same data as in Fig 4D) and PARP1 of a peptide of H3$^{1-21}$, in absence and presence of HPF1. **D:** Uncropped autoradiogram shown in Fig 4E. ADP-ribosylation reaction by PARP2-WT, PARP2-QFRD (same data as in Fig 4E) and PARP1 of histones in Nuc165, in absence and presence of HPF1. Free $^{32}$P-NAD$^+$ was run alone in the last lane as a control.
(RAR)

**S5 Fig. Quantification of the interactions within the HPF1•PARP2-WT•Nuc165 ternary complex.** FRET binding curves of HPF1_A647 to the preformed PARP2-WT_A488•Nuc165 complex (1000 nM PARP2-WT, 100 nM Nuc165) and PARP2-WT_A488 (1000 nM). Points and error bars are the mean and standard deviation from three independent measurements (no visible error bar means that the error bar is smaller than the symbol used to plot the data point). Reported $K_D$ values are the mean and standard error of the mean. All $K_D$ values are listed in Table 3.
(TIF)

**S6 Fig. Quality control of cryo-EM samples.** Samples were systematically analyzed by SDS-PAGE and 5% native PAGE after cryo-EM grid preparation. **A:** SDS-PAGE analysis of the sample that produced dataset 1. **B:** 5% native PAGE analysis of the sample that produced dataset 1. **C:** SDS-PAGE analysis of the sample that produced dataset 2. **D:** 5% native PAGE analysis of the sample that produced dataset 2.
(TIF)

**S7 Fig. Recombinant proteins and nucleosomes used in this study. A:** SDS-PAGE of 1 μg of purified recombinant PARP2-WT (lane 1), PARP2-QFRD (lane 2), Alexa488-labeled PARP2-WT (lane 3) and Alexa488-labeled PARP2-QFRD (lane 4). Proteins were detected by Coomassie staining (left panel) and covalent labeling was verified by Alexa488 fluorescence detection (right panel). **B:** SDS-PAGE of 1 μg of purified recombinant HPF1 (lane 1) and Alexa647-labeled HPF1 (lane 2). Proteins were detected by Coomassie staining (left panel) and covalent labeling was verified by Alexa647 fluorescence detection (right panel). **C:** 5% native PAGE of representative batches of Nuc165 used in this study (ethidium bromide staining). **D:** 5% native PAGE of Alexa488-labeled Nuc165 used in this study. The nucleosome was detected by ethidium bromide staining (left panel) and covalent labeling was verified by Alexa488

fluorescence detection (right panel).
(TIF)

**S8 Fig. Fluorescently labeled PARP2 and HPF1 are enzymatically active. A:** SDS-PAGE of unmodified and PARylated Alexa488-labeled PARP2-WT (Alexa488 fluorescence detection). **B:** Autoradiogram of an ADP-ribosylation reaction by PARP1 of a peptide of H3[1-21], in absence and presence of HPF1. Several representative batches of HPF1 were tested, as well as Alexa647-labeled HPF1.
(TIF)

**S1 Table.**
(TIF)

**S2 Table.**
(TIF)

## Acknowledgments

We thank Pamela N. Dyer for preparation of the human histone octamer, Alison E. White for preparation of the 165 bp 601 DNA, Annette H. Erbse for assistance with SEC-MALS experiments, Cynthia L. Page, Eileen T. O'Toole and Garry P. Morgan for assistance with cryo-EM data collection at the CU Boulder electron microscopy facility and Hui-Ting Chou for microscope operation and cryo-EM data collection at the HHMI Janelia Cryo-EM Facility. We also thank our colleagues in the Luger lab for discussion.

## Author Contributions

**Conceptualization:** Karolin Luger.

**Data curation:** Guillaume Gaullier, Genevieve Roberts, Uma M. Muthurajan, Johannes Rudolph.

**Formal analysis:** Guillaume Gaullier, Johannes Rudolph.

**Funding acquisition:** Karolin Luger.

**Investigation:** Guillaume Gaullier, Genevieve Roberts, Uma M. Muthurajan, Jyothi Mahadevan, Asmita Jha, Purushka S. Rae.

**Methodology:** Guillaume Gaullier, Genevieve Roberts, Samuel Bowerman, Johannes Rudolph.

**Project administration:** Karolin Luger.

**Supervision:** Karolin Luger.

**Validation:** Uma M. Muthurajan, Samuel Bowerman, Karolin Luger.

**Writing – original draft:** Guillaume Gaullier.

**Writing – review & editing:** Uma M. Muthurajan, Johannes Rudolph, Karolin Luger.

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
