## [Decision Letter · Decision Letter 0]

14 Sep 2020

PONE-D-20-24696

Bridging of nucleosome-proximal DNA double-strand breaks by PARP2 enhances its interaction with HPF1

PLOS ONE

Dear Dr. Luger,

Thank you for submitting your manuscript to PLOS ONE. After careful consideration, we feel that it has merit but does not fully meet PLOS ONE’s publication criteria as it currently stands. Therefore, we invite you to submit a revised version of the manuscript that addresses the points raised during the review process.

As can be seen in the reviewer comments, there are several small but important points that are best addressed. I would hope that the authors can address the concerns raised by both reviewers. 

We look forward to receiving your revised manuscript.

Kind regards,

Robert W Sobol, PhD

Academic Editor

PLOS ONE

Journal Requirements:

2.PLOS ONE now requires that authors provide the original uncropped and unadjusted images underlying all blot or gel results reported in a submission’s figures or Supporting Information files. This policy and the journal’s other requirements for blot/gel reporting and figure preparation are described in detail at https://journals.plos.org/plosone/s/figures#loc-blot-and-gel-reporting-requirements and https://journals.plos.org/plosone/s/figures#loc-preparing-figures-from-image-files. When you submit your revised manuscript, please ensure that your figures adhere fully to these guidelines and provide the original underlying images for all blot or gel data reported in your submission. See the following link for instructions on providing the original image data: https://journals.plos.org/plosone/s/figures#loc-original-images-for-blots-and-gels.

4. Please include your tables as part of your main manuscript and remove the individual files. Please note that supplementary tables (should remain/ be uploaded) as separate "supporting information" files.

Reviewers' comments:

Reviewer's Responses to Questions

**Comments to the Author**

1. Is the manuscript technically sound, and do the data support the conclusions?

Reviewer #1: Yes

Reviewer #2: Yes

2. Has the statistical analysis been performed appropriately and rigorously? 

Reviewer #1: Yes

Reviewer #2: Yes

3. Have the authors made all data underlying the findings in their manuscript fully available?

Reviewer #1: Yes

Reviewer #2: Yes

4. Is the manuscript presented in an intelligible fashion and written in standard English?

Reviewer #1: Yes

Reviewer #2: Yes

5. Review Comments to the Author

Reviewer #1: This is a well-written and interesting manuscript using a combination of in vitro biophysical tools to probe the interactions between PARP2, HPF1 and nucleosomes. A few minor comments are below.

It would be nice to note how the PARP2-QFRD mutant was serendipitously discovered.

In figure 1B, the dimerization of the nucleosome particles suggested by the authors, is not immediately apparent from the figure. The subsequent SEC-MALS analyses are more convincing than the figure displayed in 1B.

In fig. 1D, are residues R140 and W138/ Y188 displayed? If these residues are displayed to highlight conservation, it would be nicer to see them in the context of a sequence alignment as well. Are they conserved through multiple species from bacteria, archaea to mammalian species? Also, these conserved residues were not discussed in the text (only in one of the figure legends). Why these specific residues were selected for display of conservation is not discussed.

For Figure 2, Coomassie stained gels displaying the proteins/complexes would help clarify that both proteins are indeed present in the eluting fractions and clarify some of the ambiguity seen in the MW of each eluting species.

Reviewer #2: Review of PONE-D-20-24696:

The enzyme poly(ADP-ribose) polymerase-1 (PARP1), and the related enzyme PARP2, promote single- and double-strand DNA break repair in cells, by facilitating the local remodeling of chromatin into a repair-permissive configuration. PARP inhibitors suppress the repair of DNA damaged by radio- or chemotherapy, making them valuable in cancer therapy. This naturally has spurred efforts to better understand how the PARP enzymes function in chromatin.

This manuscript presents the first cryo-EM structure of PARP2 bound to a DNA end near the edge of a nucleosome. The data throughout appear very solid, the interpretations are consistent with previous studies of PARP-2 structure and function, and the methods are clearly and thoroughly described. As such, I think this study will meaningfully advance the field.

My only substantive comment relates to how PARP2 drives formation of the ‘di-nucleosomes’ used for structural analyses. At appropriate concentrations, the capacity of PARP2 to dimerize drove formation of di-nucleosomes, in which two PARP-2 molecules serve to bridge the DNA ends of two previously separate nucleosomes. The resulting complex (i.e. one PARP2 dimer and two nucleosomes) appears surprisingly homogeneous, as if the two nucleosomes were rotationally constrained relative to one another. One is tempted to attribute this homogeneity to the PARP2 dimer but, as the authors note, they cannot rule out stabilization through interactions between histone tails in one nucleosome and DNA in the adjacent nucleosome. Unfortunately, only the WGR-domain of PARP2 was amenable to structural analyses in this study. Previous X-ray crystallographic studies (PMID: 30321391) indicate that the WGR domain interacts with a 5’-phosphate at the break. However, that domain does not mediate dimer formation. Thus, while it seems highly likely that PARP2 contributes to di-nucleosome formation, it remains unclear if inter-nucleosome interactions between histones and DNA contribute as well. If such interactions occur, they would presumably be suppressed by the acetylation of histone tails in the model nucleosomes. Knowing how back-breakingly laborious cryo-EM studies can be, I don’t see this control experiment as obligatory but, ultimately, it might prove worthwhile.

6. PLOS authors have the option to publish the peer review history of their article (what does this mean?). If published, this will include your full peer review and any attached files.

Reviewer #1: No

Reviewer #2: No

---

## [Author Response · Author response to Decision Letter 0]

20 Sep 2020

We thank the reviewers for their insightful comments. Below is our response to each of the reviewer’s points. We have also addressed editorial issues and we hope that this manuscript is now suitable for publication in Plos1.

• All raw images have been deposited in Zenodo at https://doi.org/10.5281/zenodo.3519436

• We have removed the ‘data not shown’ reference, as this is not a core part of the research presented. 

Reviewer #1: This is a well-written and interesting manuscript using a combination of in vitro biophysical tools to probe the interactions between PARP2, HPF1 and nucleosomes. A few minor comments are below.

It would be nice to note how the PARP2-QFRD mutant was serendipitously discovered.

The PARP2-QFRD mutant was designed based on HDX-MS data showing protection of a peptide encompassing residues Q112 and F113 of PARP2 when mixed with PARP1, compared to PARP2 by itself. We designed the mutations to be solvent-exposed and to change drastically the properties of amino acids at these positions while not changing the protein’s isoelectric point. We edited the manuscript to explain this on page 4.

In figure 1B, the dimerization of the nucleosome particles suggested by the authors, is not immediately apparent from the figure. The subsequent SEC-MALS analyses are more convincing than the figure displayed in 1B.

We added a cartoon depiction to make our interpretation of the micrographs more explicit.

In fig. 1D, are residues R140 and W138/ Y188 displayed? If these residues are displayed to highlight conservation, it would be nicer to see them in the context of a sequence alignment as well. Are they conserved through multiple species from bacteria, archaea to mammalian species? Also, these conserved residues were not discussed in the text (only in one of the figure legends). Why these specific residues were selected for display of conservation is not discussed.

Residues W138, R140 and Y188 are not shown in figure 1D. We assume the reviewer is referring to figure S1K, to which we added the sequence alignment that was used to color the structure by conservation. The sequences used for this conservation analysis are listed in supplementary information, but we agree with the reviewer that sequence identifiers alone are not enough information, so we added to this list the species these sequences originate from. All of the PARP2 orthologues and paralogues used in this analysis are from eukaryotes, covering animals, plants and fungi (bacteria and archaea don’t have PARP enzymes). We have included an exhaustive alignment to indicate that the residues Q112, F113 are variable across species. The W138, G139 (the latter not visible in stick representation because Gly has no side chain) and R140 residues form the WGR motif that is highly conserved in DNA-dependent PARP enzymes and after which the WGR domain is named. These are denoted in the alignments as the “WGR domain”. The Y188 residue makes a key protein-DNA interaction with the terminal 5’-phosphate at a DNA break (which is an essential part of break recognition, see PMID 30321391 and PDB 6F5B) and is also highly conserved. The Y188 residue is also indicated with an arrow in the alignments. We chose to display these conserved residues to contrast with residues 112 and 113 that are not particularly conserved and to make the point that the mutations Q112R and F113D are therefore not likely to perturb the protein beyond the self-association effect we observed with the wild type protein (figures 1A, 1B and S1A). We edited the manuscript to clarify this point. 

For Figure 2, Coomassie stained gels displaying the proteins/complexes would help clarify that both proteins are indeed present in the eluting fractions and clarify some of the ambiguity seen in the MW of each eluting species.

We do not have an SDS-PAGE corresponding to these SEC-MALS traces, because unfortunately our SEC-MALS instrument is not equipped with a fraction collector.

Reviewer #2: 

The enzyme poly(ADP-ribose) polymerase-1 (PARP1), and the related enzyme PARP2, promote single- and double-strand DNA break repair in cells, by facilitating the local remodeling of chromatin into a repair-permissive configuration. PARP inhibitors suppress the repair of DNA damaged by radio- or chemotherapy, making them valuable in cancer therapy. This naturally has spurred efforts to better understand how the PARP enzymes function in chromatin.

This manuscript presents the first cryo-EM structure of PARP2 bound to a DNA end near the edge of a nucleosome. The data throughout appear very solid, the interpretations are consistent with previous studies of PARP-2 structure and function, and the methods are clearly and thoroughly described. As such, I think this study will meaningfully advance the field.

My only substantive comment relates to how PARP2 drives formation of the ‘di-nucleosomes’ used for structural analyses. At appropriate concentrations, the capacity of PARP2 to dimerize drove formation of di-nucleosomes, in which two PARP-2 molecules serve to bridge the DNA ends of two previously separate nucleosomes. The resulting complex (i.e. one PARP2 dimer and two nucleosomes) appears surprisingly homogeneous, as if the two nucleosomes were rotationally constrained relative to one another. One is tempted to attribute this homogeneity to the PARP2 dimer but, as the authors note, they cannot rule out stabilization through interactions between histone tails in one nucleosome and DNA in the adjacent nucleosome. Unfortunately, only the WGR-domain of PARP2 was amenable to structural analyses in this study. Previous X-ray crystallographic studies (PMID: 30321391) indicate that the WGR domain interacts with a 5’-phosphate at the break. However, that domain does not mediate dimer formation. Thus, while it seems highly likely that PARP2 contributes to di-nucleosome formation, it remains unclear if inter-nucleosome interactions between histones and DNA contribute as well. If such interactions occur, they would presumably be suppressed by the acetylation of histone tails in the model nucleosomes. Knowing how back-breakingly laborious cryo-EM studies can be, I don’t see this control experiment as obligatory but, ultimately, it might prove worthwhile.

In the previously published crystal structure of the WGR of PARP2 bound to DNA (PDB 6F5B, PMID 30321391, each copy of the WGR domain makes protein-DNA interactions not only with the 5’-phosphate at the break site, but also with DNA strands on both sides of the break, while the two WGR domains are too far apart to make protein-protein interactions between each other. What drives DNA bridging is therefore not a propensity of PARP2 to dimerize (it is monomeric in the absence of DNA, as we determined by SEC-MALS in figure S2A, in agreement with PMID 30321391), but the specific binding of its WGR domain to DNA ends. Our cryo-EM map is fully consistent with the conformation of the WGR domain and bridged DNA dimer present in PDB 6F5B. We speculate that the basic N-terminal region of PARP2 provides further binding affinity for DNA and could stabilize the bridged nucleosome dimer, but the resolution of our cryo-EM map is not sufficient to either support or reject this hypothesis. We appreciate the reviewer’s understanding of the amount of work that went into this cryo-EM study, and confirm that repeating it with chemically acetylated histone tails is not reasonably feasible in the timeframe we are allowed for the revisions. Additionally, in light of the recent publication from the Ahel / Matic lab (PMID 32939087) that was released last week, this is not warranted.

---

## [Decision Letter · Decision Letter 1]

6 Oct 2020

Bridging of nucleosome-proximal DNA double-strand breaks by PARP2 enhances its interaction with HPF1

PONE-D-20-24696R1

Dear Dr. Luger,

It was a pleasure to read this paper. 

We’re pleased to inform you that your manuscript has been judged scientifically suitable for publication and will be formally accepted for publication once it meets all outstanding technical requirements.

Kind regards,

Robert W Sobol, PhD

Academic Editor

PLOS ONE

Additional Editor Comments (optional):

Reviewers' comments:

Reviewer's Responses to Questions

**Comments to the Author**

1. If the authors have adequately addressed your comments raised in a previous round of review and you feel that this manuscript is now acceptable for publication, you may indicate that here to bypass the “Comments to the Author” section, enter your conflict of interest statement in the “Confidential to Editor” section, and submit your "Accept" recommendation.

Reviewer #1: All comments have been addressed

Reviewer #2: All comments have been addressed

2. Is the manuscript technically sound, and do the data support the conclusions?

Reviewer #1: Yes

Reviewer #2: Yes

3. Has the statistical analysis been performed appropriately and rigorously? 

Reviewer #1: Yes

Reviewer #2: Yes

4. Have the authors made all data underlying the findings in their manuscript fully available?

Reviewer #1: Yes

Reviewer #2: Yes

5. Is the manuscript presented in an intelligible fashion and written in standard English?

Reviewer #1: Yes

Reviewer #2: Yes

6. Review Comments to the Author

Reviewer #1: The authors have addressed all the concerns raised during the review process. No further comments at this time.

Reviewer #2: (No Response)

7. PLOS authors have the option to publish the peer review history of their article (what does this mean?). If published, this will include your full peer review and any attached files.

Reviewer #1: No

Reviewer #2: No

---

## [Editor Report · Acceptance letter]

12 Oct 2020

PONE-D-20-24696R1 

Bridging of nucleosome-proximal DNA double-strand breaks by PARP2 enhances its interaction with HPF1 

Dear Dr. Luger:

I'm pleased to inform you that your manuscript has been deemed suitable for publication in PLOS ONE. Congratulations! Your manuscript is now with our production department. 

Kind regards, 

on behalf of

Dr. Robert W Sobol 

Academic Editor

PLOS ONE